# Cryo-EM structures demonstrate human IMPDH2 filament assembly tunes allosteric regulation

Matthew C Johnson, Justin M Kollman*

Department of Biochemistry, University of Washington, Seattle, United States

**Abstract** Inosine monophosphate dehydrogenase (IMPDH) mediates the first committed step in guanine nucleotide biosynthesis and plays important roles in cellular proliferation and the immune response. IMPDH reversibly polymerizes in cells and tissues in response to changes in metabolic demand. Self-assembly of metabolic enzymes is increasingly recognized as a general mechanism for regulating activity, typically by stabilizing specific conformations of an enzyme, but the regulatory role of IMPDH filaments has remained unclear. Here, we report a series of human IMPDH2 cryo-EM structures in both active and inactive conformations. The structures define the mechanism of filament assembly, and reveal how filament-dependent allosteric regulation of IMPDH2 makes the enzyme less sensitive to feedback inhibition, explaining why assembly occurs under physiological conditions that require expansion of guanine nucleotide pools. Tuning sensitivity to an allosteric inhibitor distinguishes IMPDH from other metabolic filaments, and highlights the diversity of regulatory outcomes that can emerge from self-assembly.

*For correspondence:
jkoll@uw.edu

Competing interests: The authors declare that no competing interests exist.

## Introduction

Ribonucleotides play a central role in cellular physiology, and complex regulatory networks maintain optimal nucleotide levels according to the variable metabolic state of the cell (*Lane and Fan, 2015*). Under most conditions, cells rely on salvage pathways to regenerate degradation products and maintain nucleotide pools. However when nucleotide demand is high, for example during cellular proliferation, flux through de novo nucleotide biosynthesis pathways is up-regulated.

The enzyme IMP dehydrogenase (IMPDH) catalyzes the first committed step in guanine nucleotide synthesis. Initiation of purine nucleotide biosynthesis is tightly regulated by downstream adenine and guanine nucleotide products. Balancing the flux through these parallel synthesis pathways, which share the precursor inosine monophosphate (IMP), is essential for cellular homeostasis (*Figure 1A*) (*Allison and Eugui, 2000*). IMPDH is regulated transcriptionally, post-translationally, and allosterically (*Hedstrom, 2009*). In vertebrates, two IMPDH isoforms, (83% identical in humans), have differential expression patterns (*Collart and Huberman, 1988*; *Natsumeda et al., 1990*). IMPDH1 is constitutively expressed at low levels in most tissues, while IMPDH2 is generally upregulated in proliferating tissues (*Carr et al., 1993*; *Hager et al., 1995*; *Jackson et al., 1975*; *Senda and Natsumeda, 1994*). In mice, knockout of IMPDH1 results in only very minor vision defects, whereas knockout of IMPDH2 is embryonic lethal (*Aherne et al., 2004*; *Gu et al., 2003*; *Gu et al., 2000*).

IMPDH reversibly assembles into filaments in vertebrate cells and tissues, which is thought to provide an additional layer of regulation (*Carcamo et al., 2011*; *Ji et al., 2006*; *Thomas et al., 2012*). Self-assembly into filamentous polymers has recently been observed in cells for a large number of metabolic enzymes (*Narayanaswamy et al., 2009*; *Noree et al., 2019*; *Noree et al., 2010*; *O'Connell et al., 2014*; *Shen et al., 2016*; *Werner et al., 2009*). Filament assembly is typically induced by changes in metabolic conditions, and the enzymes involved are key regulatory or branch point enzymes, supporting the hypothesis that self-assembly is a general mechanism of regulating

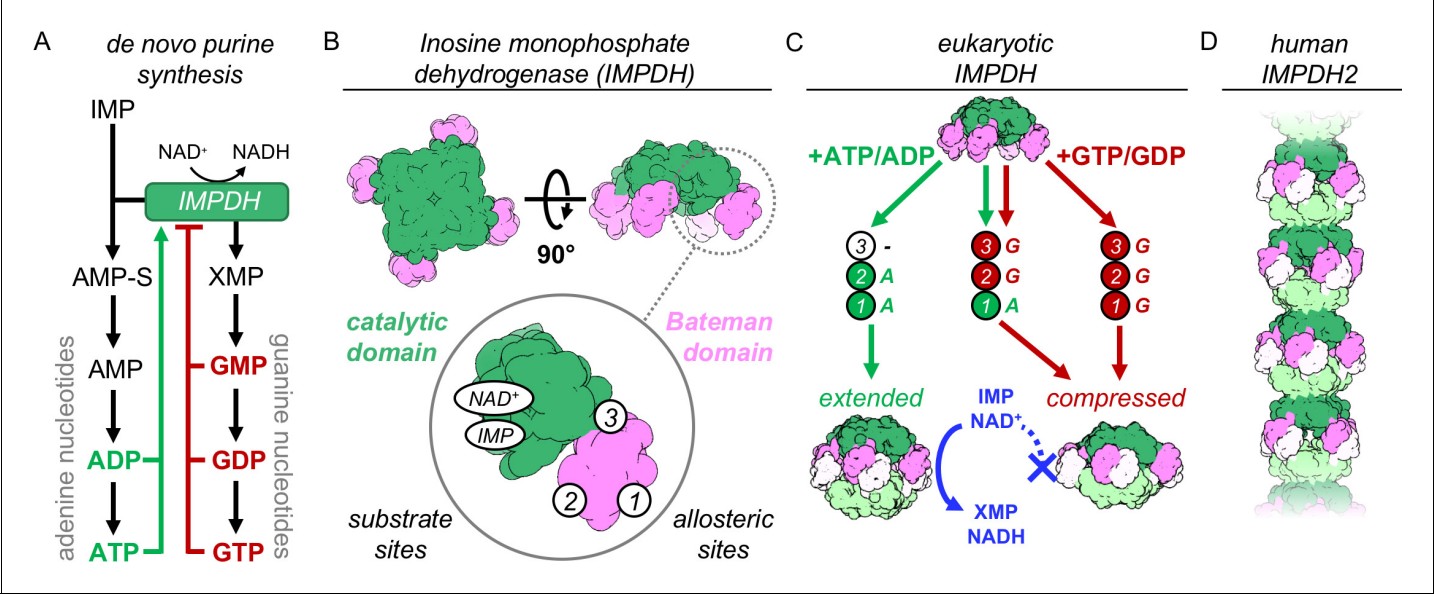

**Figure 1.** IMPDH structure and function. (**A**) De novo purine nucleotide biosynthesis pathways. (**B**) IMPDH consists of a catalytic domain with two substrate binding sites (green), and a regulatory Bateman domain (pink) with three allosteric binding sites on the Bateman domain numbered 1,2,3. (**C**) Bound nucleotides promote regulatory domain dimerization, forming reversible IMPDH octamers that may be active or inhibited. Opposing tetramers colored light green and light pink. (**D**) Human IMPDH2 assembles into filaments composed of canonical octamers.

flux through specific metabolic pathways (*Aughey and Liu, 2016*). In the few cases where the structure and function of metabolic filaments have been described, they show that the polymers directly influence enzyme activity by stabilizing specific active or inhibited conformations (*Hunkeler et al., 2018*; *Kim et al., 2019*; *Lynch et al., 2017*; *Lynch and Kollman, 2020*; *Stoddard et al., 2019*; *Webb et al., 2018*). Low resolution structural studies have shown that IMPDH2 filaments are flexible, adopting both active and inhibited conformations, and that filament assembly does not directly affect enzyme activity (*Anthony et al., 2017*; *Labesse et al., 2013*).

Guanine deprivation induces assembly of IMPDH into micron-scale ultrastructures composed of bundled filaments, which disassemble once homeostasis is restored (*Calise et al., 2014*; *Juda et al., 2014*; *Labesse et al., 2013*; *Thomas et al., 2012*). Additional stimuli that alter IMPDH catalytic flux result in assembly, including increased intracellular IMP, and treatment with IMPDH inhibitors or other anti-proliferative drugs (*Chang et al., 2015*; *Ji et al., 2006*; *Keppeke et al., 2015*; *Keppeke et al., 2018*). In vivo, IMPDH assembly is seen mainly in cases of high nucleotide demand, for example assembly is correlated with proliferation of mouse induced pluripotent stem cells and human T-cell activation (*Calise et al., 2018*; *Duong-Ly et al., 2018*; *Keppeke et al., 2018*). However, the molecular mechanisms that underlie assembly-dependent regulation of IMPDH have not been described.

The catalytic mechanism and the structures of IMPDH monomers and defined oligomers are well-described *Hedstrom (2009)*. IMPDH forms stable tetramers, each protomer consisting of a catalytic domain and a regulatory Bateman domain (*Figure 1B*). The enzyme converts IMP to xanthosine monophosphate (XMP) and requires NAD$^+$ as a cofactor for a complex multi-stage catalysis requiring rearrangements of active site loops that close around the substrate upon binding, and then must reopen to allow release of product. The regulatory domain contains three allosteric sites that bind adenine and guanine nucleotides (*Buey et al., 2015b*). Sites 1 and 2 are canonical cystathionine beta synthase motifs that bind either ATP/ADP or GTP/GDP, and are conserved among IMPDH homologues (*Baykov et al., 2011*; *Ereño-Orbea et al., 2013*; *Ignoul and Eggermont, 2005*; *Scott et al., 2004*). Site 3 is a non-canonical site located at the interface between domains that binds only GTP/GDP (*Buey et al., 2015b*).

In eukaryotes, adenine and guanine nucleotides allosterically modulate IMPDH activity by altering oligomeric state (*Buey et al., 2017*; *Fernández-Justel et al., 2019*). IMPDH tetramers reversibly assemble into octamers when nucleotides bind the two canonical sites and drive dimerization of

Bateman domains (*Figure 1C*). Bound adenines promote an extended conformation in which the active sites are free to open and close as needed for catalysis. GTP/GDP binding induces a compressed conformation by changing the relative orientation of the two domains. This brings the active sites of opposing tetramers tightly together, forming an interdigitated pseudo beta-barrel that prevents reopening and product release and inhibits IMPDH activity by a mechanism described as a 'conformational switch' between extended and compressed states (*Buey et al., 2017*; *Buey et al., 2015b*; *Fernández-Justel et al., 2019*). Because inhibition requires interactions between two opposing tetramers, this switch is functionally relevant in only the octameric state.

In vitro treatment with ATP or GTP induces assembly of human IMPDH into filaments composed of stacked octamers interacting through their catalytic domains (*Figure 1D*) (*Anthony et al., 2017*; *Fernández-Justel et al., 2019*; *Labesse et al., 2013*). Filament segments can both extend and compress, and assembly does not have a direct effect on the activity of IMPDH2 (*Anthony et al., 2017*). Importantly, mutations that block assembly of IMPDH filaments in vitro also prevent assembly of the large IMPDH bundles observed in cells, supporting the functional relevance of in vitro reconstituted filaments.

In this study, we present a series of cryo-electron microscopy (cryo-EM) structures of human IMPDH2. Structures of the enzyme treated with multiple combinations of substrates (IMP, NAD+) and allosteric effectors (ATP, GTP), in both filament and non-filament assembly states, demonstrate the extreme conformational plasticity of the enzyme. These structures define the interactions that drive filament assembly, and we show that in vitro IMPDH2 filament assembly is sensitive to the same conditions that promote assembly in cells: high IMP levels and low guanine nucleotide levels (*Keppeke et al., 2018*). Finally, we show that filament assembly tunes sensitivity to GTP inhibition by stabilizing a conformation that reduces affinity for GTP. In other enzyme polymers with known functions, assembly directly up or down regulates activity; the resistance to feedback inhibition observed in IMPDH2 filaments demonstrates a new role for enzyme self-assembly and reflects the diversity of allosteric regulatory mechanisms in metabolic filaments.

From these data, we have developed a model (described in more detail below), in which IMPDH2 filament assembly modulates known conformational changes of the enzyme to alter catalytic flux in response to proliferative signaling. Eukaryotic IMPDH activity is regulated in response to adenine or guanine nucleotides by extension or compression, respectively, of the Bateman domains. Under physiologically high substrate (IMP) levels, assembled filaments resist the compressed, inhibited, state, independent of guanine levels. This allows for a third regulatory state in which the enzyme is able to resist feedback inhibition in response to proliferative signaling.

## Results

### IMPDH2 filaments are conformationally heterogeneous

We first characterized IMPDH2 filaments assembled in vitro by addition of ATP (*Anthony et al., 2017*). The affinity of IMPDH2 for ATP has not been directly measured but we found ATP concentrations as low as 1 µM sufficient to induce assembly (*Figure 2A*). We prepared cryo-EM grids of ATP-assembled filaments and found that in the absence of other ligands, IMPDH2 filaments are extremely flexible (*Figure 2—figure supplement 1A*). Two-dimensional class averages confirmed our prior observation from negative stain that the filaments are composed of conformationally heterogeneous octamers stacked head-to-head, resulting in flexible filaments with variable rise and radius of curvature (*Figure 2B*). Because these deviations from ideal helical symmetry severely limited attempts at image processing by traditional iterative helical real-space reconstruction, we attempted to produce more structurally homogeneous filaments by addition of IMP or NAD+, which stabilize the flexible active site loops (*Sintchak et al., 1996*). As previously reported, substrates did not have a direct effect on IMPDH2 filament assembly, and filament assembly did not directly affect enzymatic activity (*Figure 2C–D*) (*Anthony et al., 2017*). While the turnover rate we observe here (~0.05 per second) is somewhat low, these reactions were performed near room temperature (24°C); we found that increasing reaction temperature to 37°C resulted in increasing Kcat to ~0.1 per second, roughly half what was observed in another recently published study of human IMPDH2 (*Figure 2—figure supplement 2*) (*Fernández-Justel et al., 2019*). Unfortunately, addition of IMP and NAD+, either alone or in combination, did not significantly reduce filament flexibility (*Figure 2—figure supplement 1B–D*).

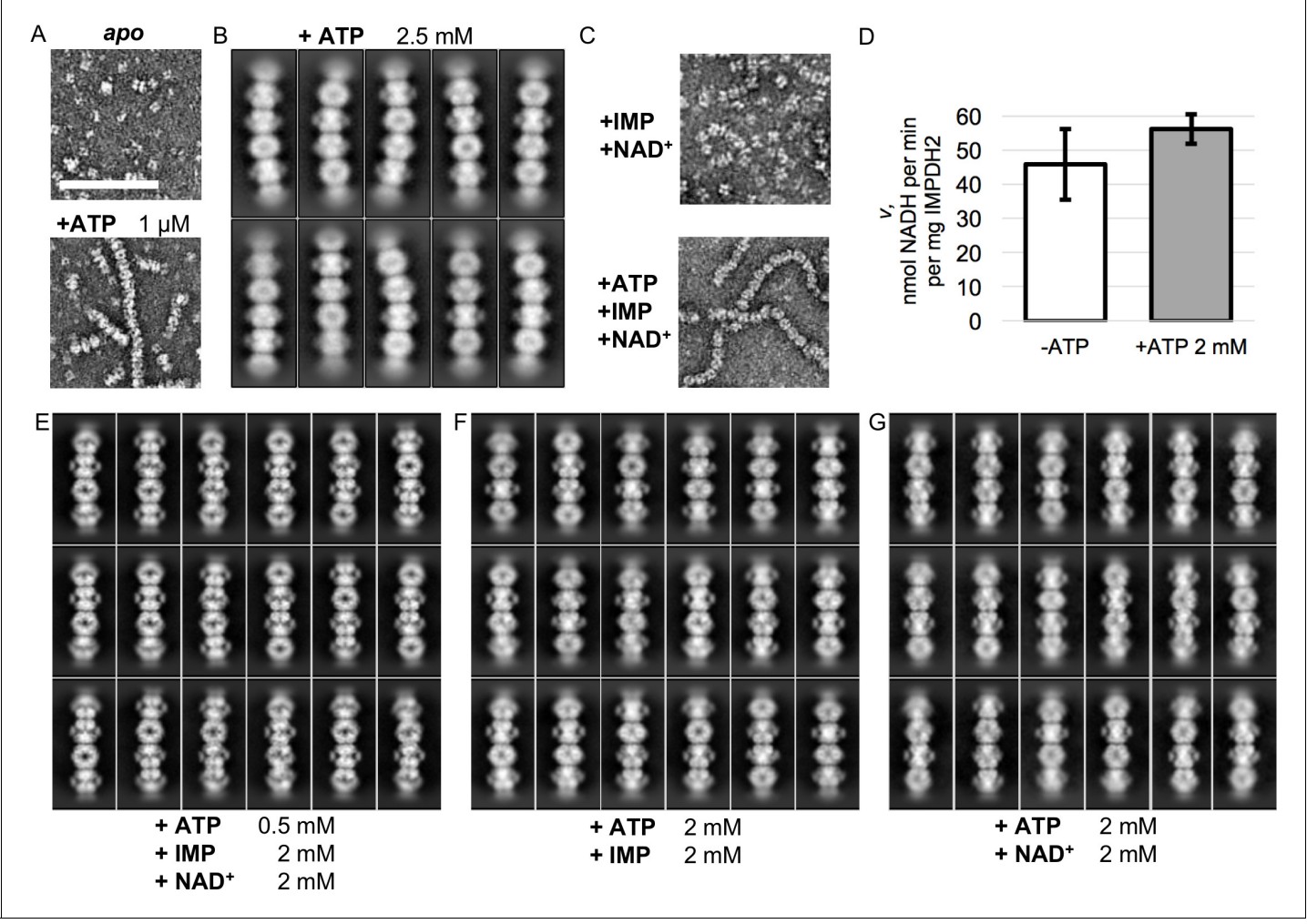

**Figure 2.** Electron microscopy of uninhibited IMPDH2 filaments. (**A**) Negative stain EM of purified human IMPDH2. Treatment with 1 μM ATP induces filament assembly. Scale bar 100 nm. (**B**) Representative 2D class averages from the 2.5 mM ATP cryo-EM dataset. (**C**) Negative stain EM of actively catalyzing IMPDH2 (2 mM IMP, 2 mM NAD$^+$), with and without 2 mM ATP. (**D**) Initial velocity of enzyme (2 mM IMP, 2 mM NAD$^+$), with and without 2 mM ATP. Average of three replicates, error bars + /- 1 s.D. (**E–G**) Representative 2D class averages from the three uninhibited enzyme cryo-EM datasets, with nucleotide concentrations as indicated at bottom.

The online version of this article includes the following figure supplement(s) for figure 2:

**Figure supplement 1.** A cryo-EM image processing workflow for structure determination of flexible IMPDH2 filaments.

**Figure supplement 2.** Kinetic data of IMPDH2.

Two-dimensional class averages of helical segments again exhibited varying degrees of curvature and showed no correlation of structural states between neighboring IMPDH2 octamers, preventing successful three-dimensional processing with conventional helical approaches (*Figure 2E–G*).

From these 2D class averages we observed that filament flexibility was due to variations in the conformation of filament segments, but that the interface between segments did not vary, and as a consequence this region was better resolved (*Figure 3A*). Focused refinement of the filament assembly interface region alone provided a valuable foothold in resolving high resolution structures of all regions of the filaments. We developed a workflow for a single-particle style approach to reconstruction of inherently heterogeneous helical protomers, combining density subtraction, symmetry expansion, focused classification and focused refinement to isolate, classify, and reconstruct discrete sections of filaments for structural characterization (*Figure 2—figure supplement 1E–G*). This approach provided multiple structures for two different focused regions of IMPDH2 filaments: full

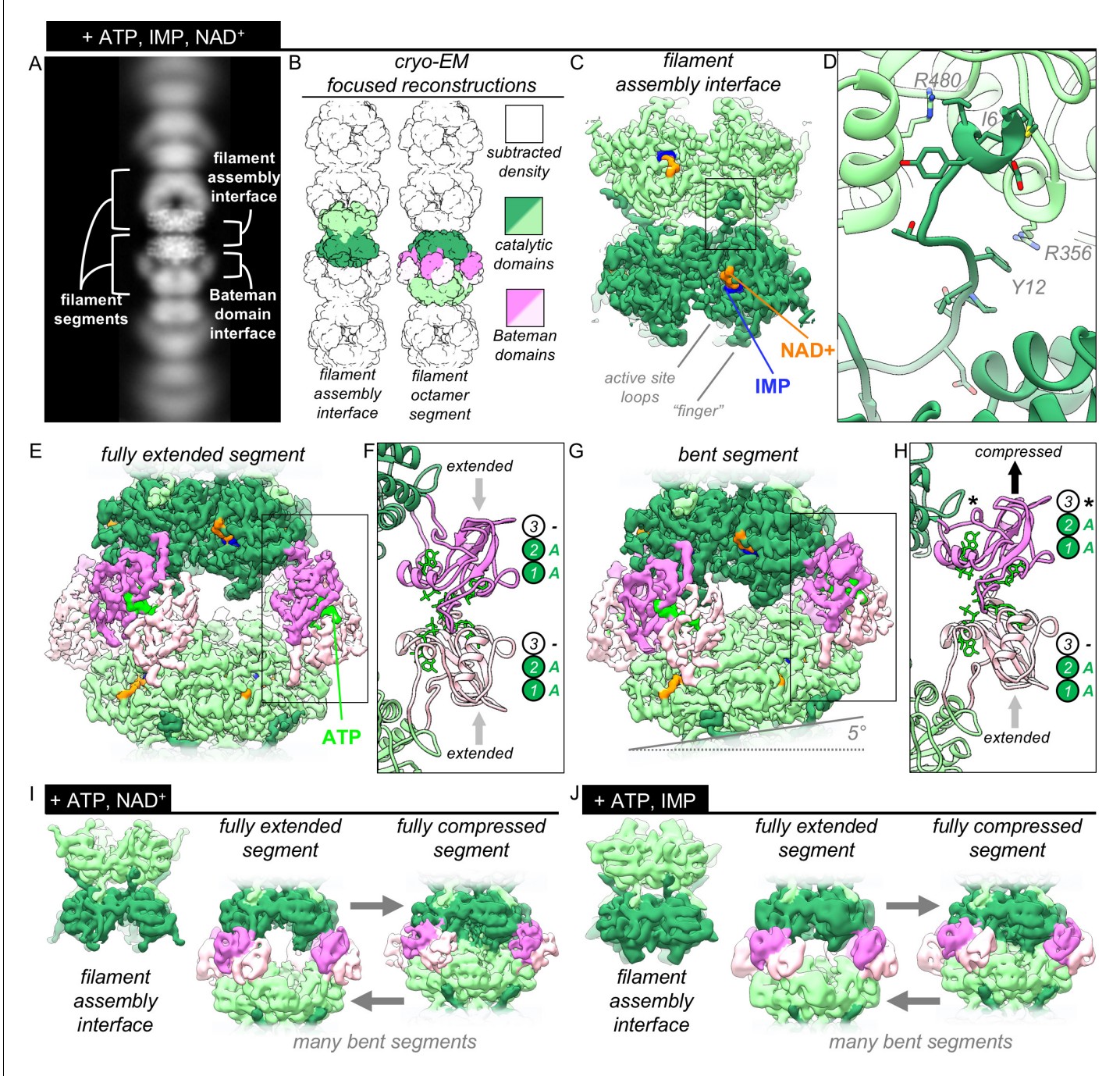

**Figure 3.** The structures of uninhibited IMPDH2 filaments. (**A**) Cryo-EM of IMPDH2 filaments with both substrates (representative 2D class average). (**B**) We resolved two types of structures from IMPDH filaments: the consensus filament assembly interface, and various conformations of filament segments. (**C**) Cryo-EM density for the ATP/IMP/NAD+ consensus filament assembly interface, consisting of two tetramers bound back-to-back (dark and light green) (0.5 mM ATP, 2 mM IMP, 2 mM NAD⁺). (**D**) The filament assembly interface is mediated by the vertebrate-specific N-terminus, in particular a key bridge between Y12 and R356. (**E**) Cryo-EM density for the ATP/IMP/NAD+ fully extended filament segment. Opposing catalytic tetramers (dark and light green), are held separate by their symmetrically extended Bateman domains (dark and light pink). ATP (bright green) is resolvable in the Bateman domains. (**F**) In the fully extended Bateman domains, sites 1 and 2 are occupied by ATP, and site 3 is unformed. (**G**) Cryo-EM density for the best resolved ATP/IMP/NAD+ bent filament segment, in which the two catalytic tetramers are not parallel. (**H**) Filament segment bending results from asymmetric compression of Bateman domains. In this reconstruction, one protomer from each of the two tetramers is compressed, and allosteric site 3 is formed, but unoccupied (black asterisk). (**I**) Summary of the ATP/NAD⁺ cryo-EM dataset (2 mM ATP, 2 mM NAD⁺). The filament assembly interface is unchanged, and filament segments varied from fully extended, to bent, to fully compressed. In the absence of IMP, the flexible active site loops are disordered. (**J**) Summary of the ATP/IMP cryo-EM dataset (2 mM ATP, 3 mM IMP).

*Figure 3 continued on next page*

*Figure 3 continued*

The online version of this article includes the following source data and figure supplement(s) for figure 3:

**Source data 1.** Statistics of cryo-EM data collection, reconstruction and model refinement for the ATP/IMP/NAD$^+$ dataset.
**Source data 2.** Statistics of cryo-EM data collection, reconstruction and model refinement for the ATP/NAD$^+$ dataset.
**Source data 3.** Statistics of cryo-EM data collection, reconstruction and model refinement for the ATP/IMP dataset.
**Figure supplement 1.** Image processing of the IMPDH2 +ATP, IMP, NAD$^+$ cryo-EM dataset.
**Figure supplement 2.** Model/Map FSC curves for the IMPDH2 +ATP, IMP, NAD$^+$ cryo-EM dataset.
**Figure supplement 3.** The vertebrate-specific N-terminus mediates IMPDH2 assembly of ATP-bound IMPDH2 filaments, in which individual protomers can extend or compress freely.

octameric segments, and the paired catalytic domain tetramers that constitute the filament assembly interface (*Figure 3B*).

## The IMPDH2 filament assembly interface is well-ordered

To define the interactions that drive assembly of catalytically active IMPDH2 filaments, we obtained cryo-EM structures of IMPDH2 in three liganded states: 1) ATP/IMP/NAD$^+$, 2) ATP/NAD$^+$, and 3) ATP/IMP. The three datasets were qualitatively similar. However, because the structures refined to higher resolution when both substrates were present, we focus our analysis here on the ATP/IMP/NAD$^+$ dataset. The complete image processing workflow applied to this dataset (*Figure 3—figure supplement 1A*) was also applied to the two single-substrate datasets. From the ATP/IMP/NAD$^+$ data we resolved many different structural classes, three best-resolved of which were: the filament assembly interface, and two different reconstructions of octameric filament segments (*Figure 3— source data 1*). We resolved many conformations of filament segments, however for all segments the filament assembly interface was identical. Therefore the filament assembly interface structure is a dataset consensus structure, an average of every segment included in the dataset. We did not observe any conformational cooperativity in the compressed/extended conformational equilibrium between IMPDH2 octamers sharing a filament assembly interface.

We determined the structure of the ATP/IMP/NAD$^+$ consensus filament assembly interface at 3.0 Å resolution (*Figure 3C*, *Figure 3—figure supplement 1B–C*, *Figure 3—figure supplement 2A*). This D4 symmetric region is composed of two symmetrically opposed catalytic domain tetramers. The interface between tetramers is formed by the 12 amino-terminal residues of eight protomers, which each extend from the core of the molecule to bind one catalytic domain on the opposite tetramer, in a shallow surface groove formed by a short helix (476-485), two beta strands (51-63), and two short loops (355-360, 379-380) (*Figure 3D*). A key tyrosine/arginine interaction (Y12/R356') anchors the attachment; mutation of either of these residues to alanine was previously shown to abolish assembly, both in vivo and in vitro (*Anthony et al., 2017*). Residues 1–7 make mostly hydrophobic interactions with the catalytic domain, and an embedded arginine (R480') is positioned to hydrogen bond to the I6 backbone carbonyl. These interactions are reciprocal - a protomer extends its N-terminus to a partner and receives the N-terminus of the same partner, creating four pairs of symmetrical interactions at the interface. The overall surface area buried by a single interaction is 2,300 Å$^2$, with a total buried area of 9,200 Å$^2$ for the multimeric interface.

The N-terminal tail of IMPDH, corresponding to residues 1–28 in IMPDH2, is the least conserved part of the protein, with large variation in length and sequence between phyla, as well as conformational variation among known structures (*Figure 3—figure supplement 3A*) (*Buey et al., 2015a*; *Fernández-Justel et al., 2019*; *Kim et al., 2017*; *Labesse et al., 2013*; *Makowska-Grzyska et al., 2015*; *Osipiuk et al., 2014*; *Prosise and Luecke, 2003*; *Trapero et al., 2018*). The residues involved in filament contacts, however, are conserved among chordates, consistent with the fact that IMPDH polymerization has only been reported in vertebrates (*Figure 3—figure supplement 3B*). In previous crystal structures of human and fungal IMPDH, residues 1–11 are unresolved and residues 12–28 are well resolved. In these structures, Asp16 anchors the tail in place through ionic interactions with Arg341/Lys349 of a neighboring protomer, while Val13 and Pro14 make hydrophobic contacts. In the filament structure, however, the Val13/Pro14 contacts are broken and the entire tail is rotated about 30° to position Tyr12 to contact Arg356 across the filament interface (*Figure 3—figure supplement 3C*).

In this consensus structure, the remainder of the catalytic domain is nearly identical to a structure of human hIMPDH2 bound to competitive inhibitors in a previously determined crystal structure (PDB 1nf7, backbone RMSD 0.641 Å for residues 18–107, 245–417, and 439–514) (*Sintchak et al., 1996*). The active site is well-resolved, except for one loop (residues 421–436). There is strong density in the both the IMP and NAD$^+$ binding sites. We have modeled these ligands as IMP and NAD$^+$, however because the enzyme filaments were actively turning over when flash frozen, we have likely captured a mixture of substrate-, intermediate-, and product-bound states. Attempts at focused classification of the active site to structurally isolate these states were unsuccessful.

## The extended conformation is an ensemble of flexible states

We resolved multiple conformations of filament segments in the ATP/IMP/NAD$^+$ dataset (*Figure 3—figure supplement 1D*). The most well-resolved was a D4-symmetric, fully extended octamer (3.3 Å, *Figure 3E*, *Figure 3—figure supplement 1E–H*, *Figure 3—figure supplement 2B*). The eight Bateman domains are symmetrically extended with respect to their catalytic domains, giving the octameric segment a helical rise of 118 Å; a symmetric helix of these segments would have a twist of 36˚. There is strong density at both of the canonical ATP-binding sites within the Bateman domain (*Figure 3F*), consistent with previous crystal structures of ATP/ADP-bound fungal and bacterial IMPDH (*Buey et al., 2017*; *Labesse et al., 2013*).

We also resolved several bent filament segment structures, the best resolved of which reached 3.9 Å (*Figure 3G*, *Figure 3—figure supplement 1I–L*, *Figure 3—figure supplement 2C*). This bent octamer contains two identical tetramers. For each, three protomers are in extended conformations, while one protomer is in the compressed conformation. Due to the symmetry of the octamer, the lone compressed protomer of each tetramer forms a Bateman domain dimer with an extended protomer of the opposing tetramer (*Figure 3H*, *Figure 3—figure supplement 3D*). From each of the lower resolution ATP/IMP and ATP/NAD$^+$ datasets, we also observed a single consensus filament assembly interface and many different filament segment classes, including fully extended and bent segments, as well as fully compressed segments (*Figure 3I–J*, *Figure 3—source data 2*, *Figure 3—source data 3*). To our knowledge, these are the first structure of IMPDH in the compressed conformation in the absence of guanine nucleotides. While the two canonical ATP sites are occupied, the third Bateman binding site (which forms only in the compressed state and is GTP/GDP-specific) remains unoccupied. This suggests that protomers within ATP-bound IMPDH filaments readily sample the compressed conformation, and that GTP binding to site 3 selectively stabilizes this state. Further, the ability of a compressed protomer to form a Bateman dimer with an extended partner demonstrates a lack of conformational cooperativity across the Bateman domain interface.

## The balance of substrate and downstream product regulates filament assembly

We and others previously reported that GTP can stabilize compressed IMPDH2 filaments or drive their disassembly, depending on what other ligands are present (*Anthony et al., 2017*; *Buey et al. (2017)*; *Duong-Ly et al., 2018*). To understand how ligand status of the active site tunes the response to GTP, we systematically explored the effects of different ligand combinations on filament assembly. In the absence of substrate, GTP induces disassembly of filaments, but as little as 10 µM IMP inhibits disassembly (*Figure 4A–B*). Pre-treatment of filaments with IMP prevents disassembly by GTP, and addition of IMP promotes reassembly of filaments previously disassembled by GTP (*Figure 4—figure supplement 1A–B*).

To understand how IMP and GTP allosterically influence filament assembly and disassembly of ATP-bound IMPDH2, we acquired cryo-EM data of the enzyme in two ligand states: 1) ATP/GTP/IMP, and 2) ATP/GTP. To ensure morphological consistency, we sought to saturate the enzyme with GTP. For the ATP/GTP dataset, we used 2 mM GTP, which for both our initial negative stain experiments and cryo-EM preparations resulted in complete disassembly of filaments into free octamers (*Figure 4—figure supplement 1C*). However, under saturating IMP concentrations, 2 mM GTP resulted in filaments that were often bent (*Figure 4—figure supplement 1D*). This should not be possible if GTP were saturating all sites; our structures above suggest bent filaments must contain some extended protomers whose GTP-binding allosteric site 3 is disrupted. For the ATP/GTP/IMP cryo-EM dataset we therefore used a much higher GTP concentration (20 mM), which resulted in

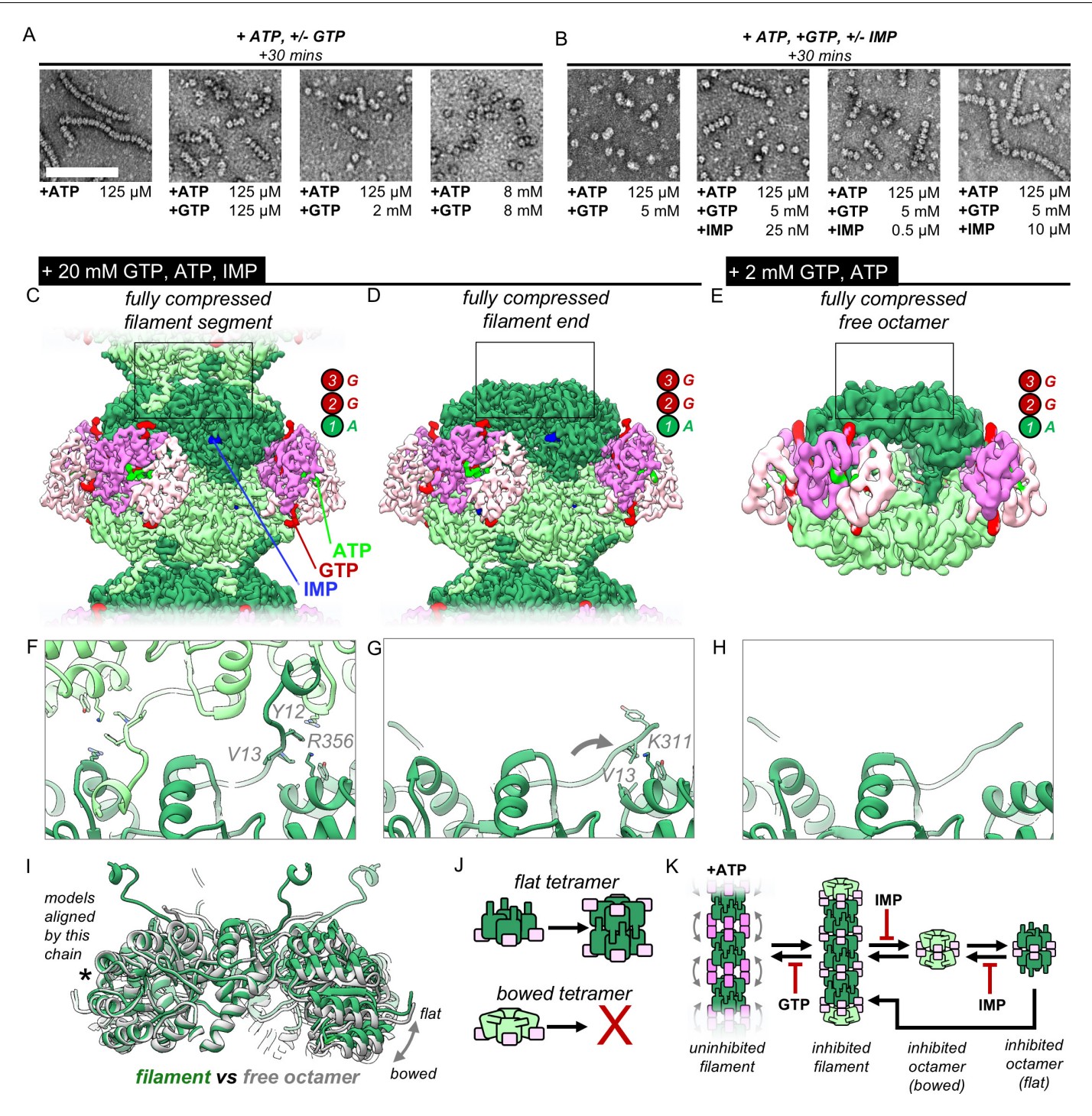

**Figure 4.** IMP and GTP allosterically modulate filament assembly and disassembly. (A) Roughly 2 mM GTP inhibits filament assembly of IMPDH by ATP. Negative stain EM, protein concentration 2 uM. Scale bar 100 nm. Final nucleotide concentrations for each EM grid were as indicated below each image. (B) Roughly 10 uM IMP inhibits filament assembly by GTP. Final nucleotide concentrations for each EM grid were as indicated below each image. (C) Composite cryo-EM density of the GTP/ATP/IMP filament assembly interface and fully compressed filament segment maps (20 mM GTP, 0.5 mM ATP, 1 mM IMP). (D) Cryo-EM density of the fully compressed filament end map. (E) Cryo-EM density of the GTP/ATP non-filament fully compressed free octamer map (2 mM GTP, 2 mM ATP). (F–H) Close-up ribbon views of the assembled and unassembled filament interfaces the maps in A-C. (I) Comparison between the tetramer conformations of the 'flat' assembled filament interface (green) and the 'bowed' unassembled free octamer (gray). (J) Cartoon of the relationship between tetramer bowing and filament assembly. (K) Model of the regulation of filament assembly by GTP and IMP.

*Figure 4 continued on next page*

*Figure 4 continued*

The online version of this article includes the following source data and figure supplement(s) for figure 4:

**Source data 1.** Statistics of cryo-EM data collection, reconstruction and model refinement for the ATP, 2 mM GTP dataset.
**Source data 2.** Statistics of cryo-EM data collection, reconstruction and model refinement for the ATP/IMP, 20 mM GTP dataset.
**Figure supplement 1.** Electron microscopy of human IMPDH2 treated with ATP, GTP, IMP, and NAD$^+$.
**Figure supplement 2.** Image processing of the IMPDH2 +ATP, IMP, 20 mM GTP cryo-EM dataset.
**Figure supplement 3.** Model/Map FSC curves for the IMPDH2 +ATP, IMP, 20 mM GTP cryo-EM dataset.
**Figure supplement 4.** Image processing of the IMPDH2 +ATP, 2 mM GTP cryo-EM dataset.
**Figure supplement 5.** Model/Map FSC curves for the IMPDH2 +ATP, 2 mM GTP cryo-EM dataset.
**Figure supplement 6.** The assembled IMPDH2 filament interface is not compatible with the 'bowed' tetramer conformation seen in the unassembled GTP-bound free octamer.

filaments with fully compressed segments (*Figure 4—figure supplement 1E*). We explore this apparent difference in GTP affinity in more detail below.

The compressed filament dataset resulted in three high-resolution reconstructions: the consensus filament assembly interface, a fully compressed octameric filament segment, and a fully compressed octameric end segment (*Figure 4—figure supplement 2A*, *Figure 4—source data 1*). The ATP/GTP/IMP filament assembly interface map (3.0 Å) is identical (backbone RMSD = 0.407) to the ATP/IMP/NAD$^+$ filament interface (*Figure 4—figure supplement 2B–C*, *Figure 4—figure supplement 3A*). Classifying the most well-resolved filament segments as before, we obtained a 3.2 Å structure of a fully compressed filament segment (*Figure 4C*, *Figure 4—figure supplement 2D–H*, *Figure 4—figure supplement 3B*). The Bateman domains are symmetrically compressed, with ligand density at all three allosteric sites. The active site is well resolved, including the canonical interactions between opposing active site fingers that inhibit catalysis by preventing catalytic dynamics (*Buey et al., 2017*).

Compared to the uninhibited filament datasets, these GTP-saturated, +IMP filaments were shorter in length. As a result, in addition to filament segments, we identified many filament ends: octamers in which one tetramer does not have an assembled interface. The best-resolved structure of these filament ends (3.3 Å) is conformationally similar to the filament segments, being fully compressed, with well-resolved IMP-bound active sites in the inhibited conformation; however, without the filament assembly interface, the N-terminus is only partially resolved (*Figure 4D*, *Figure 4—figure supplement 2I–L*, *Figure 4—figure supplement 3C*).

From the free octamer dataset containing ATP/GTP, we used a similar symmetry-expansion and classification strategy to that used for the filament datasets (*Figure 4—figure supplement 4A*). This scheme confirmed that virtually all the particles were symmetric, fully compressed free octamers in which both potential filament assembly interfaces are unbound, with some poorly resolved minority classes of larger oligomers (*Figure 4—figure supplement 4B–C*). The best resolved free octamer class reached intermediate resolution (4.5 Å) (*Figure 4E*, *Figure 4—figure supplement 4D–F*, *Figure 4—figure supplement 5*, *Figure 4—source data 2*). This octamer is conformationally similar (backbone RMSD = 0.978 Å) to a recent higher-resolution crystal structure of IMPDH2 bound to GTP (PDB 6i0o) (*Buey et al., 2017*; *Fernández-Justel et al., 2019*). The Bateman domains are fully compressed with three occupied allosteric sites. Without IMP present, the active site loops of the free octamers are disordered, with partial density for the fingers extending across to the opposing tetramer.

Whether in a filament segment, filament end, or free octamer, the Bateman domains of the three ATP/GTP-bound IMPDH octamer types are fully compressed, with ligand density in all regulatory sites including the critical GTP/GDP-specific third site, which 'staples' IMPDH octamers in the fully compressed, inhibited conformation. But we observed two key structural differences that correlated with assembly state: the conformation of the N-terminus, and the relative orientation of protomers within each tetramer. The filament interface of the inhibited segments is unchanged from the uninhibited filaments (*Figure 4F*). However, at the free filament ends the N-terminus is only partially unresolved (a.a. 1–11), with the resolvable portion rotated ~30° degrees compared to the bound

interface, such that Val13 inserts into the shallow hydrophobic pocket formed by A307, A308, and Lys311 of the neighboring protomer, very similar to its position in the GTP-bound free octamer crystal structure (PDB 6i0o) (*Figure 4G*) (*Fernández-Justel et al., 2019*). The resolution of the free octamer structure precludes side chain placement, but the backbone density of the unbound N-terminus is in the same position, indicating that free filament ends and free octamers are in the same conformation (*Figure 4H*).

We noticed a striking difference between the conformation of the catalytic core assembly in free versus assembled states. Each protomer of the free octamer and at filament ends is tilted ~5° relative to the four-fold symmetry axis, such that the tetramer becomes more 'bowed' than the 'flat' tetramers at filament assembly interfaces (*Figure 4I*, *Video 1*). The tetramers found in the GTP-bound free octamer crystal structure (PDB 6i0o) are in a similar bowed conformation (*Fernández-Justel et al., 2019*). The bowed tetramer conformation and the filament assembly interface are mutually exclusive. When the N-terminus conformation from the free octamer is modeled into the flat tetramer of the filament assembly interface, V13 is not correctly positioned to bind the A307/A308/K311 of the neighboring protomer, and instead clashes with K311 and A307 (*Figure 4—figure supplement 6A*). Thus the flat conformation promotes release of the N-termini from their binding sites on neighboring protomers, freeing them to rotate into the conformation seen in the assembled filament interface. The reciprocal operation is also not possible; for a bowed tetramer modeled with the N-terminus conformation from the filaments, only one protomer at a time can form the assembled interface (*Figure 4—figure supplement 6B*). The other N-termini are out of position, with significant separation of the critical residues Y12 and R356 as well as multiple steric clashes. Thus bowed tetramers cannot form the IMPDH2 filament assembly interface, and tetramer flattening is a necessary precondition for IMPDH2 filament assembly (*Figure 4J*).

These three structures of inhibited IMPDH2 conformations provide a model by which filament assembly is influenced by IMP and GTP through tetramer bowing and flattening (*Figure 4K*). In the absence of substrate, GTP induces both compression of filament segments and tetramer bowing, with the latter resulting in filament disassembly. But when IMP is bound, the disordered active site loops become ordered and rigid, buttressing intra-tetramer contacts as well as forming a pseudo-beta-barrel between opposing tetramers in the GTP-bound state. These increased contacts work to resist tetramer bowing and more readily sample the flat tetramer conformation, which promotes, and is stabilized by, the filament assembly interface. When IMP levels are low, GTP promotes filament disassembly, but high IMP levels shift the equilibrium towards filament assembly.

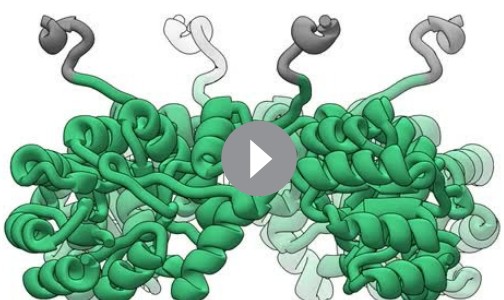

**Video 1.** Comparison between the 'flat' tetramer of assembled filaments and the 'bowed' tetramer of free octamers. Morph comparison between catalytic tetramers of the GTP/ATP/IMP filament assembly interface and the GTP/ATP free octamer. For visualization purposes, we have depicted the complete N-terminus in both conformations, however in the free octamer model, some residues were unresolved (gray). For both models, opposing tetramers, Bateman domains, and active site loops have been hidden from view.
https://elifesciences.org/articles/53243#video1

## IMPDH2 filaments resist GTP inhibition

Based on this model, during quiescence, when the salvage pathways supply ample GTP and IMP production is downregulated, intracellular IMPDH2 will be in the fully compressed, fully inhibited, ATP/GTP-bound free octamer state. Without increased IMP production, guanine depletion will result in transient octamer extension and filament assembly, both of which will reverse as the resulting increase in IMPDH flux restores guanine levels by diverting substrate from the parallel de novo adenine pathway, mirroring the extension/compression behavior of non-filament assembling homologues

(*Buey et al., 2017*; *Fernández-Justel et al., 2019*). Recently, it was shown that in vivo IMPDH assembly is promoted by increases in intracellular IMP (*Keppeke et al., 2018*). We therefore reasoned that the primary regulatory function of filament assembly may apply only in the proliferative state, when intracellular IMP levels are upregulated, preventing filament disassembly by GTP.

To probe whether filament assembly influences the regulatory effects of GTP on IMPDH2 activity we compared enzyme kinetics of the wildtype enzyme with the non-assembling mutant Y12A. We measured the GTP dose-response of IMPDH2 pre-incubated with varying levels of ATP, and found that wild-type enzyme is less sensitive to GTP inhibition, compared with Y12A (*Figure 5—figure supplement 1A–C*). Depending on ATP concentration, the apparent GTP IC50 of the wild-type was roughly two-fold lower than for Y12A. At higher ATP levels, the apparent GTP IC50 of WT and the non-assembly mutant both increase. We attribute this to competition between ATP and GTP at the first and second Bateman sites, suggesting that independent of filament assembly, GTP inhibition of IMPDH2 is affected by ATP. Notably, the range of GTP in which the substrate-saturated filaments resist GTP inhibition is within the upper range of in vivo concentrations (*Traut, 1994*). One function, then, of IMPDH2 filaments is to resist inhibition by GTP.

To correlate structural differences in IMPDH2 filaments as a function of GTP concentration, we collected negative stain EM data directly from three reaction volumes used for the GTP inhibition experiments, corresponding to uninhibited filaments (no GTP),~10% inhibited filaments (2.5 mM GTP), and fully inhibited filaments (20 mM GTP) (*Figure 5A*). As in the cryo-EM datasets, we found that uninhibited filaments were often extended, and fully inhibited filaments were universally compressed. However, we noted that the partially inhibited filaments contained a more heterogeneous mix of extended, bent, and compressed segments.

Next, we compared NAD$^+$ kinetics of uninhibited (0 mM GTP) and partially inhibited (2 mM GTP) filaments with saturating IMP concentrations (*Figure 5B–C*). As expected, in the absence of GTP WT and the non-assembling mutant have similar apparent Michaelis-Menten kinetics (*Anthony et al., 2017*). Inhibition of WT by this concentration of GTP is partial and non-competitive. In contrast, the non-assembling mutant is strongly inhibited. In the absence of saturating IMP, these same GTP levels result in complete octamer compression and filament disassembly, providing a possible mechanism by which filament assembly alters GTP inhibition: by resisting the fully compressed state.

To better understand this partially inhibited state, we collected cryo-EM data of IMPDH2 with saturating substrates, 0.5 mM ATP, and 2 mM GTP (*Figure 5—figure supplement 1D*). Under these conditions, we observed the non-assembly mutant Y12A was 84% inhibited but WT enzyme was only 16% inhibited. We resolved structures of not only the filament assembly interface and several distinct filament segments, but surprisingly, two different free octamers; a canonical free octamer (a dimer of tetramers bound by Bateman domains), and also a small class of free interfacial octamers (a dimer of tetramers bound by the filament assembly interface) (*Figure 5—figure supplement 1E*, *Figure 5—figure supplement 2A*, *Figure 5—source data 1*).

The 3.1 Å resolution consensus assembly interface map is identical to the uninhibited and fully inhibited consensus interfaces, including a well-resolved active site with strong density for both substrates; as with the ATP/IMP/NAD+ structures these filaments are actively turning over and we have likely captured many states, which we have modeled simply as IMP/NAD$^+$ (*Figure 5—figure supplement 2B–C*, *Figure 5—figure supplement 5A*, *Figure 5—figure supplement 7A*). As expected, the filament segments exhibited a range of conformations (*Figure 5—figure supplement 2D*). The best resolved of these were a fully compressed filament segment at 3.4 Å resolution, and two bent conformations at 4.2 and 3.7 Å resolution (*Figure 5—figure supplement 2E–M*, *Figure 5—figure supplement 5B–D*). The fully compressed filament segment is identical to the corresponding structure from the inhibited (20 mM GTP) filament dataset, with all Bateman binding sites occupied (*Figure 5—figure supplement 7B*). As with the bent conformation from the ATP/IMP/NAD$^+$ dataset, the two bent filament segments are each two-fold symmetric octamers of asymmetric tetramers containing different proportions of extended or compressed protomers (*Figure 5—figure supplement 7C–D*). The asymmetric unit from one of these bent segments is a tetramer with two compressed and two extended protomers, and the other has three compressed and one extended. For each of the compressed protomers, there is clear ligand density bound at Bateman Site 3; for the extended protomers this site is unformed and empty.

Unlike the mixture of extended and compressed protomers we observed in the filaments, the free canonical octamers were universally compressed (*Figure 5—figure supplement 3A–C*). As with

the ATP/GTP free octamer, the best resolved canonical free octamer class (3.8 Å) is fully compressed and bowed, despite having both substrate sites occupied (*Figure 5—figure supplement 3D–F*, *Figure 5—figure supplement 6A*, *Figure 5—figure supplement 7E*). Thus, even in the IMP-bound state, the compressed/flat conformation is less preferred, unless stabilized by the filament assembly interface. The small class of free interfacial octamers (3.8 Å) was no different from the filament interface maps, except that the Bateman domains are completely unresolved, indicating that without the stabilization provided by Bateman domain dimerization, these regions are highly flexible (*Figure 5—figure supplement 4A–F*, *Figure 5—figure supplement 6B*, *Figure 5—figure supplement 7F*). The observation of dramatically different structural ensembles in filament-bound and free IMPDH2, in the presence of identical ligand concentrations, explains the role of filament assembly in resisting compression and GTP inhibition.

## Filament-specific IMPDH2 conformations reduce GTP affinity and promote activity

From our different cryo-EM datasets combined, we have now determined structures of canonical IMPDH2 octamers bound to allosteric effectors and both substrates, in six distinct conformations (*Figure 5D*). From the uninhibited ATP/IMP/NAD$^+$ dataset we resolved a fully extended filament segment, and a bent segment in which for each tetramer, 3 protomers were extended and 1 was compressed. From the partially inhibited GTP/ATP/IMP/NAD$^+$ dataset, we resolved 2:2, 3:1, and fully compressed filament segments, and a fully compressed free octamer. These five structures provide a mechanism by which the Bateman domain extension of discrete protomers promoted by filament assembly resists GTP inhibition (*Figure 5E*). Depending on the degree of extension/bending/compression, filament segments exhibit a range of increasingly stronger interactions between opposing catalytic domains. For the fully extended segments, there are no interactions, and the flexible active loops are able to perform the complex conformational changes necessary for catalysis (*Buey et al., 2017*; *Hedstrom, 2009*). Going through the progressively more compressed bent states, there is progressively greater surface area buried by a series of distinct contacts between the opposing active site loops, until the fully compressed filament segment. For the active sites that make these contacts, activity is likely impaired because the active site loops are constrained. However, the presence of some unconstrained active sites in the bent filament segments means that these protomers are catalytically active. This effect is not cooperative within the octamers, and even a single extended protomer is sufficient to reduce inhibitory active site contacts.

The Bateman domain ligand occupancy of these five filament segment conformations varies significantly. Given the resolution range of these structures (3–4 Å) it is not possible to unequivocally distinguish ATP from GTP, and we have assigned ligand identity according to previous structures of a bacterial IMPDH bound to ATP (PDB 4dqw) (*Labesse et al., 2013*) and a fungal IMPDH bound to ATP and GDP (PDB 5tc3) (*Buey et al., 2017*). These were chosen due to conformational similarity of the Bateman domains of these structures to our ATP/IMP/NAD$^+$ fully extended and ATP/IMP/GTP fully compressed human filament cryo-EM structures; backbone RMSDs were 1.163 and 0.856 Å, respectively (for residues corresponding to human IMPDH2 residues 110–244). As previously described, in both the fully extended and 1:3 compressed:extended segments there is clear ligand density at Bateman sites 1 and 2 (*Figure 5F*). For the single compressed protomer in the latter, site 3 is formed but due to the absence of GTP in that dataset it is unoccupied. In the partially inhibited filament dataset, for which the buffer contained both ATP and GTP, we see a greater variation in Bateman ligand occupancy. For the 2:2 compressed:extended segment, there is full occupancy at sites 1, 2, and 3 in the compressed protomers, however for the two extended protomers there is strong density at site 1 and partial density at site 2. The 3:1 compressed:extended segment is qualitatively similar; the three compressed protomers possess full ligand occupancy but for the one extended protomer site 2 has only partial density. For both the fully compressed filament segment and the free octamer there is full ligand density. Thus filament assembly promotes the extended state of individual protomers, which reduces overall GTP affinity due to disruption of site 3.

To probe whether the conformations of filament segments co-vary along the length of the filaments, we subjected the five filament-interface cryo-EM datasets presented here to classification against a series of synthetic templates with different conformations, and compared the conformations of interfaced octamers through calculations of odds ratios (*Figure 5—figure supplements 8–13*). The results of the classification against synthetic templates are in line with the experimentally

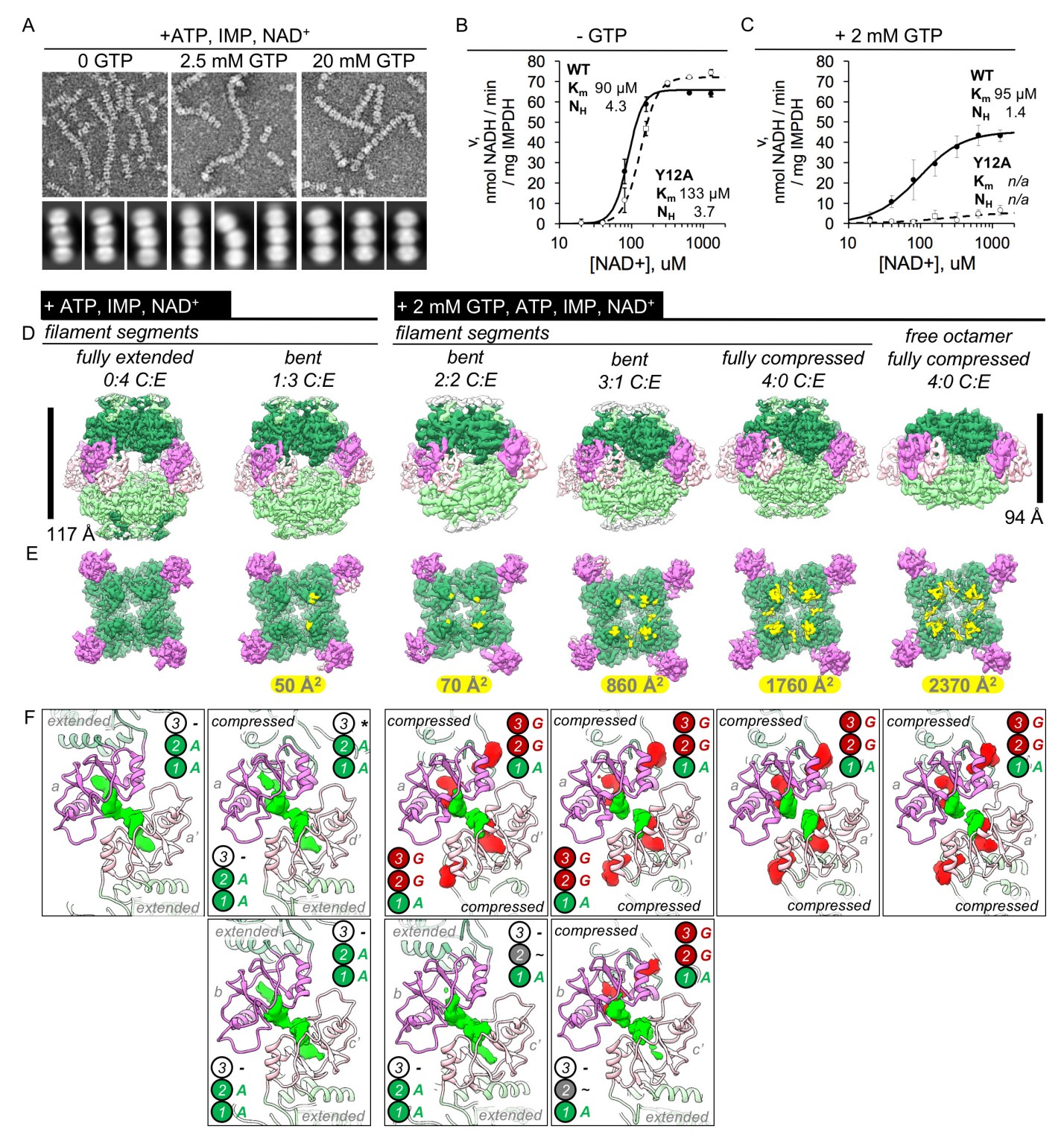

**Figure 5.** IMPDH2 filaments resist GTP inhibition by promoting bent octamer conformations that separate opposing active sites. (**A**) Negatively stained EM of uninhibited (left), partially inhibited (center), and fully inhibited (right) IMPDH2. Representative micrographs and reference free 2D class averages. Prepared with 1 mM IMP, 1 mM NAD$^+$, and either 0, 2.5 mM, or 20 mM GTP. (**B–C**) NAD+ saturation curves of uninhibited WT IMPDH2 (solid line), and the non-assembly mutant Y12A (dashed line). Reactions performed with 0.5 mM ATP, 1 mM IMP, and varying NAD$^+$. (**C**) NAD+ saturation curves of WT filaments treated with 0.5 mM ATP, 1 mM IMP, varying NAD$^+$ and 2 mM GTP. (**D**) Six cryo-EM maps from two datasets (uninhibited ATP/IMP/NAD+ and partially inhibited ATP/IMP/NAD+/[2 mM]GTP) exhibiting a range of Bateman domain conformations. The uninhibited filaments were prepared

*Figure 5 continued on next page*

*Figure 5 continued*

with 0.5 mM ATP, 2 mM IMP, and 2 mM NAD$^+$. The partially inhibited filaments were prepared with 0.5 mM ATP, 2 mM IMP, and 2 mM NAD$^+$, and 2 mM GTP. (E) A view of a single tetramer from the inside of each octamer. The lighter colored tetramer from panel A is hidden, with the surface area buried between tetramer active sites colored in yellow, with the indicated total buried surface area. (F) Corresponding views of Bateman domain conformations. Protein displayed as ribbon, with the two interacting Bateman domains colored orchid and light pink. Cryo-EM density for non-protein ligand densities is colored green and red, for ATP and GTP, respectively. Symmetry identities labeled with gray letters. In the extended conformation, allosteric site 3 is distorted, and does not bind ligands (black dashes). Allosteric site 3 is formed in compressed protomers, but in the absence of guanine nucleotides it remains unoccupied (black asterisk). In the extended protomers of some bent octamers, is is possbile that allosteric site 2 is only partially occupied (black tilde).

The online version of this article includes the following source data and figure supplement(s) for figure 5:

**Source data 1.** Statistics of cryo-EM data collection, reconstruction and model refinement for the ATP/IMP/NAD$^+$, 2 mM GTP dataset.
**Figure supplement 1.** IMPDH2 filaments resist GTP inhibition.
**Figure supplement 2.** Image processing of the IMPDH2 +ATP, IMP, NAD$^+$, 2 mM GTP cryo-EM dataset, part 1: initial processing, and processing of filament segments.
**Figure supplement 3.** Image processing of the IMPDH2 +ATP, IMP, NAD$^+$, 2 mM GTP cryo-EM dataset, part 2: the free canonical octamers.
**Figure supplement 4.** Image processing of the IMPDH2 +ATP, IMP, NAD$^+$, 2 mM GTP cryo-EM dataset, part 3: the free interfacial octamers.
**Figure supplement 5.** Model/Map FSC curves for the IMPDH2 +ATP, IMP, NAD$^+$, 2 mM GTP cryo-EM dataset (filament structures).
**Figure supplement 6.** Model/Map FSC curves for the IMPDH2 +ATP, IMP, NAD$^+$, 2 mM GTP cryo-EM dataset (non-filament structures).
**Figure supplement 7.** Bateman domains of partially inhibited IMPDH2 filaments are in a mix of compressed and extended states.
**Figure supplement 8.** Classification of interfaced octamer pairs against synthetic templetes, and calculation of interfaced octamer odds ratios.
**Figure supplement 9.** Interfaced octamer odds ratios for the 0.5 mM ATP, 2 mM IMP, 2 mM NAD$^+$ cryo-EM dataset.
**Figure supplement 10.** Interfaced octamer odds ratios for the 2 mM ATP, 3 mM IMP cryo-EM dataset.
**Figure supplement 11.** Interfaced octamer odds ratios for the 2 ATP, 2 mM NAD$^+$ cryo-EM dataset.
**Figure supplement 12.** Interfaced octamer odds ratios for the 2 mM GTP, 0.5 mM ATP, 2 mM IMP, 2 mM NAD$^+$ cryo-EM dataset.
**Figure supplement 13.** Interfaced octamer odds ratios for the 20 mM GTP, 0.5 mM ATP, 1 mM IMP cryo-EM dataset.

derived classifications for each dataset, but we did not find a strong correlation in segment conformation along filaments, suggesting that conformational changes are not cooperative across the filament interface.

## Discussion

Understanding the complex ways cells regulate IMPDH activity to efficiently maintain spatiotemporal control of nucleotide levels in response to varying demand has direct implications for human health. IMPDH activity is upregulated to increase guanine levels in proliferating tissues like tumors and regenerating liver (*He et al., 2018*; *Huang et al., 2018*; *Nagai et al., 1991*; *Tressler et al., 1994*; *Yalowitz and Jayaram, 2000*). IMPDH plays a particularly important role in the immune response, where T-cell activation is dependant on increased production of purine nucleotides, and is associated with IMPDH filament assembly (*Calise et al., 2018*; *Duong-Ly et al., 2018*; *Gu et al., 2000*; *Zimmermann et al., 1998*). As a result, IMPDH is the target of several drugs used in immunosuppressive treatment of both autoimmune disease and organ transplant rejection, and is considered a promising target for antineoplastic agents (*Bergan et al., 2016*; *Liao et al., 2017*; *Shu and Nair, 2008*).

Assembly and disassembly of IMPDH into filaments has been observed in healthy proliferative cells, and in cancer cells (*Keppeke et al., 2018*; *Wolfe et al., 2019*). IMPDH filaments reversibly assemble in stimulated T-cells as they transition to a proliferative state, in a mechanism dependent on multiple metabolic signaling pathways and on the levels of guanine nucleotides (*Calise et al., 2018*; *Duong-Ly et al., 2018*). Despite the importance of understanding human IMPDH regulation, until now the regulatory role of IMPDH filament assembly has not been explored at the structural level.

We propose a model that describes the regulatory role of human IMPDH2 filament assembly, in which assembly reduces feedback inhibition of enzyme activity in a substrate-dependant manner, increasing flux through the de novo guanine nucleotide synthesis pathway in response to proliferative signaling. In the absence of either IMP or guanine ligands, IMPDH2 is conformationally dynamic

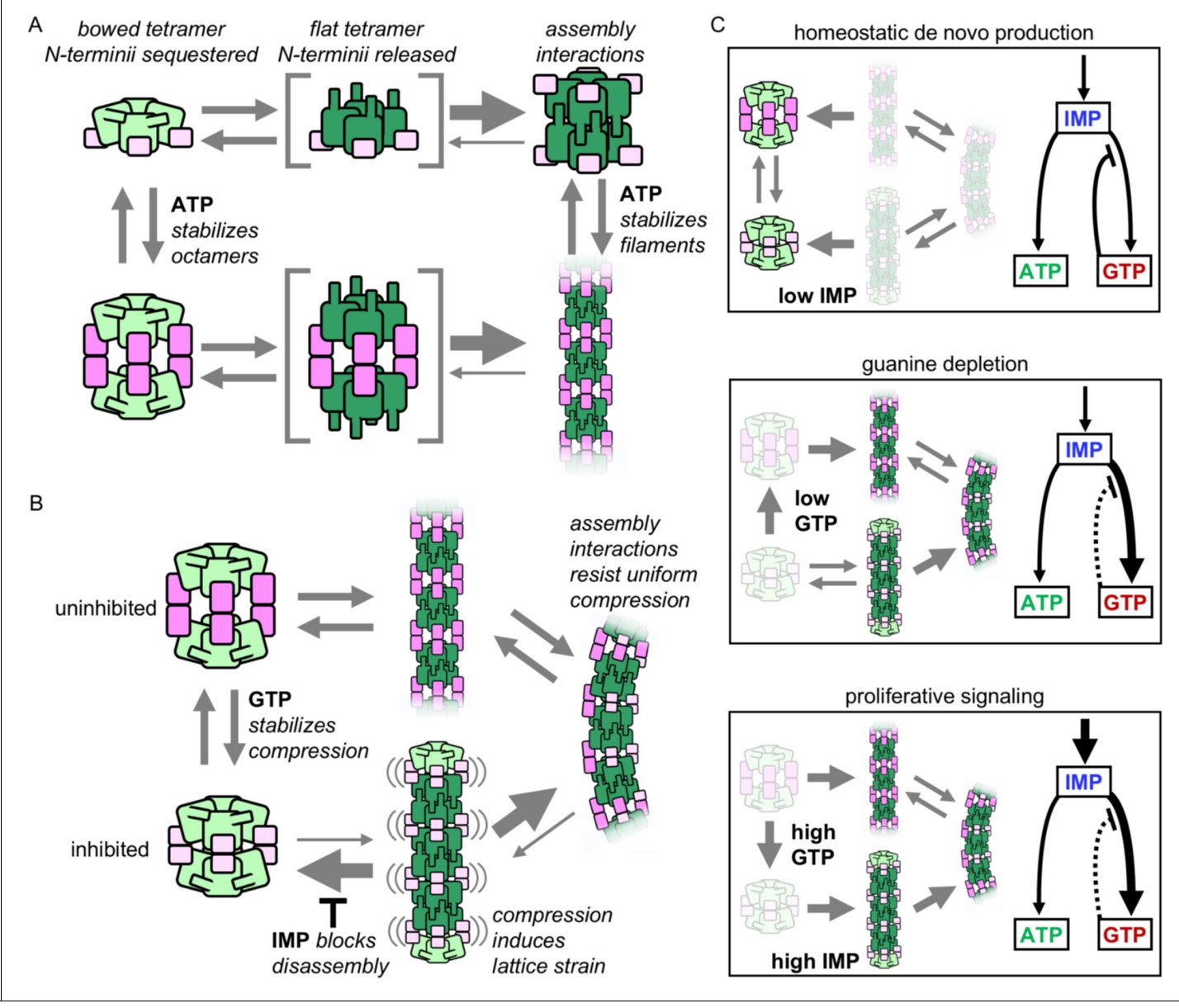

**Figure 6.** Model of IMPDH2 assembly and filaments' role in guanine nucleotide regulation. (**A**) Filament assemble when octamer interactions are stabilized by ATP binding to the regulatory domain (pink), and the N-terminal residues (blue) are released by flattening of the catalytic tetramer (green). Filament assembly interactions stabilize the flat conformation. (**B**) GTP binding stabilizes an inhibited, compressed conformation. Filament is less sensitive to GTP-induced compression, maintaining a population of octamers in mixed activity states. Filaments in the fully compressed GTP-bound state are strained, which promotes disassembly that is inhibited by substrate IMP binding. (**C**) The equilibrium in (**B**) explains the different cellular conditions in which IMPDH polymerization occurs. (i) Under homeostatic conditions IMPDH2 is dispersed and activity is regulated by GTP binding to octamers, which balances low levels of de novo synthesis between adenine and guanine pathways. ii) When guanine nucleotides are depleted the equilibrium shifts toward filaments. iii) Proliferative signaling can directly shift the equilibrium toward filaments, where higher flux is maintained through the guanine pathway under elevated GTP concentrations due to reduced sensitivity of the filaments to GTP inhibition (dashed line).

(*Figure 6A*). Apo IMPDH2 forms stable tetramers, which freely sample both the 'bowed' and 'flat' tetramer conformations, with the latter resulting in release of the N-terminus and assembly into stable interfacial octamers. Adenine nucleotides bind with high affinity to the Bateman domain, resulting in stable filaments in which the active site loops remain unconstrained and the enzyme is active.

The ATP concentration required to induce filament assembly in vitro is far below the expected in vivo levels; we thus predict the apo state to be rare in human cells (*Traut, 1994*). Without bound guanine nucleotides, the Bateman domains of individual protomers freely compress and extend (*Figure 6B*). However, the flat tetramer conformation found in assembled IMPDH2 filaments is resistant to full compression of octameric filament segments. Binding of GTP to the Bateman domain stabilizes the compressed state, leading to lattice strain that is relieved by disassembly of the filament interface and tetramer bowing. In this way high intracellular guanine levels disassemble IMPDH2 filaments into stable free octamers whose activity is inhibited. Binding of IMP to the active site stabilizes the flexible active site loops, and saturation with both IMP and GTP results in compressed filaments in which the active sites of opposing tetramers interlock into a stable network. This rigidifies the octameric filament segment, which now resists the lattice strain brought on by compression, blocking GTP-induced filament disassembly.

By balancing these complex conformational dynamics, cells fine-tune feedback inhibition of the enzyme, consistent with the states in which IMPDH filaments are observed in cells (*Figure 6C*). Under homeostatic conditions, the salvage pathways provide sufficient purine nucleotides, and IMP production is low. Under these conditions IMPDH2 is bound to both adenine and guanine nucleotides, but not IMP, forming free octamers rather than filaments. In vivo, filaments are typically not observed in quiescent cells; rather, as in our model, their assembly is associated with increased intracellular IMP and decreased intracellular GTP (*Calise et al., 2014*; *Juda et al., 2014*; *Keppeke et al., 2018*; *Schiavon et al., 2018*). Depletion of guanine allows Bateman domain extension, and transient assembly into enzymatically active filaments. However, upon restoration of guanine levels, these active filaments disassemble into compressed free octamers, mirroring the known feedback inhibition behavior of non-filament forming IMPDH homologues (*Buey et al., 2017*; *Buey et al., 2015b*; *Fernández-Justel et al., 2019*).

The key difference is the substrate dependance of IMPDH2 filament assembly. IMP-saturated IMPDH2 filaments resist both disassembly and the fully compressed, inhibited state. Even at elevated guanine nucleotide levels, these filaments retain a proportion of uninhibited active sites. This allows the cell to modulate enzyme activity to balance levels of product and substrate in response to metabolic demand, which can vary significantly depending on cell type and cell cycle stage. Our data strongly support the idea that IMPDH2 filament assembly serves to elevate intracellular guanine nucleotide levels during the proliferative state by resisting feedback inhibition. Production of IMP is upregulated in response to overall purine demand, as well as in response to proliferative signaling via the mechanistic target of rapamycin (mTOR) (*Ben-Sahra et al., 2016*; *Smith, 1998*). Inhibition of mTOR reverses IMPDH filament assembly in activated mouse T-cells, as well as proliferating mouse liver cells (*Chang et al., 2015*; *Duong-Ly et al., 2018*). Thus the dependence of filament assembly on IMP levels provides an avenue for regulation of assembly through established proliferative signaling pathways.

Many metabolic enzymes form filamentous polymers in cells in response to changes in metabolic state (*Ingerson-Mahar et al., 2010*; *Narayanaswamy et al., 2009*; *Noree et al., 2010*; *O'Connell et al., 2012*; *Petrovska et al., 2014*; *Saad et al., 2017*; *Shen et al., 2016*; *Zhao et al., 2013*). Most of these metabolic filaments are assembled from important regulatory enzymes, which suggests polymerization may play a role in modulating flux through these pathways. Recently, in just a few cases, structural and biochemical studies have provided insight into the functional consequences of enzyme filament assembly, suggesting that one role of polymers is to regulate activity by locking enzymes into active or inactive conformations through assembly contacts (*Barry et al., 2014*; *Hunkeler et al., 2018*; *Lynch et al., 2017*; *Stoddard et al., 2019*; *Webb et al., 2018*). Thus, it was surprising when our initial characterization of IMPDH2 filaments showed that assembly did not affect enzymatic activity or the ability to switch between active or inactive conformations (*Anthony et al., 2017*). Instead, as we have shown here, IMPDH2 filaments fine-tune the the allosteric response by reducing affinity for inhibitory downstream products. This represents a new way in which enzyme self-assembly can modulate flux through metabolic pathways, providing an additional layer of regulatory control on top of existing transcriptional, post-translational, and allosteric regulation.

# Materials and methods

**Key resources table**

| Reagent type (species) or resource | Designation | Source or reference | Identifiers | Additional information |
|---|---|---|---|---|
| Strain, strain background (*Eschericia coli*) | BL21(DE3) | Thermo Scientific | EC0114 | competent cells |
| Recombinant DNA reagent | pSMT3-hIMPDH2-WT (pJK053) | https://doi.org/10.1091/mbc.E17-04-0263 | | expression plasmid |
| Recombinant DNA reagent | pSMT3-hIMPDH2-Y12A (pJK061) | https://doi.org/10.1091/mbc.E17-04-0263 | | expression plasmid |
| Peptide, recombinant protein | Ubiquitin-like-specific protease 1 (ULP1) | https://doi.org/10.1091/mbc.E17-04-0263 | | purified in house |
| Peptide, recombinant protein | human IMPDH2 | https://doi.org/10.1091/mbc.E17-04-0263 | | purified in house |
| Chemical compound, drug | LB broth mix | LabExpress | 3003 | bacterial growth media |
| Chemical compound, drug | IPTG | GoldBio | I2481C100 | |
| Chemical compound, drug | MgCl2 | Fisher Scientific | BP215-500 | |
| Chemical compound, drug | KPO4 | Fisher Scientific | BP362-500 | |
| Chemical compound, drug | KCl | Fisher Scientific | BP217-3 | |
| Chemical compound, drug | imidazole | Sigma Aldrich | SLBT7469 | |
| Chemical compound, drug | urea | Fisher Scientific | BP169-212 | |
| Chemical compound, drug | DTT | Fisher Scientific | 172–25 | |
| Chemical compound, drug | HEPES | Fisher Scientific | BP310-1 | |
| Chemical compound, drug | ATP | Sigma Aldrich | A2383-10G | |
| Chemical compound, drug | GTP | Sigma Aldrich | G8877-1G | |
| Chemical compound, drug | IMP | Sigma Aldrich | 57510–5G | |
| Chemical compound, drug | NAD+ | Sigma Aldrich | N6522-1G | |

*Continued on next page*

*Continued*

| Reagent type (species) or resource | Designation | Source or reference | Identifiers | Additional information |
|---|---|---|---|---|
| Chemical compound, drug | uranyl formate | Electron Microscopy Sciences | 22450 | negative stain EM |
| Software, algorithm | GCTF | https://doi.org/10.1016/j.jsb.2015.11.003 | | |
| Software, algorithm | Relion | https://doi.org/10.7554/eLife.42166 | | |
| Software, algorithm | MotionCor2 | https://doi.org/10.1038/nmeth.4193 | | |
| Software, algorithm | SWISS-MODEL | https://doi.org/10.1093/nar/gky427 | | |
| Software, algorithm | UCSF Chimera | https://doi.org/10.1002/jcc.20084 | | |
| Software, algorithm | COOT | https://doi.org/10.1107/S0907444910007493 | | |
| Software, algorithm | PHENIX | https://doi.org/10.1107/97809553602060000865 | | |
| Software, algorithm | LocScale | https://doi.org/10.7554/eLife.27131 | | |
| Software, algorithm | Molprobity | https://doi.org/10.1107/S0907444909042073 | | |
| Software, algorithm | EMRinger | https://doi.org/10.1038/nmeth.3541 | | |
| Other | C-flat 2/2 holey carbon films on Cu 200 mesh grid | Protochips, Inc | CF-2/2–2C | cryo-EM sample preparation |
| Other | HisTrap FF Crude 5 ml | GE Life Sciences | 17528601 | protein purification |
| Other | Superose 6 Increase 10/300 GL | GE Life Sciences | 29-0916-96 | protein purification |
| Other | Amicon Ulta-15 30K MWCO centrifugal filters | Millipore | UFC903008 | protein concentration |

## Recombinant IMPDH expression and purification

Purified hIMPDH2 was prepared as described previously (*Anthony et al., 2017*). BL21 (DE3) *E. coli* transformed with a pSMT3-Kan vector expressing N-terminal SMT3/SUMO-tagged hIMPDH2 were cultured in Luria broth at 37°C until reaching an $OD_{600}$ of 0.8 and then induced with 1 mM IPTG for 4 hr at 30°C and pelleted. The remainder of purification was performed at 4°C. Pellets were resuspended in lysis buffer (50 mM KPO4, 300 mM KCl, 20 mM imidazole, 800 mM urea, pH 8) and lysed with an Emulsiflex-05 homogenizer. Lysate was cleared by centrifugation and SUMO-tagged hIMPDH2 chromatagraphically purified with HisTrap FF columns (GE Healthcare Life Sciences) and an Äkta Start chromatography system. After in-column washing with lysis buffer and elution (with 50 mM KPO4, 300 mM KCl, 500 mM imidazole, pH 8), peak fractions were treated with 1 mg ULP 1 protease (*Mossessova and Lima, 2000*) per 100 mg hIMPDH2 for 1 hr, followed by the addition of 1 mM dithiothreitol (DTT) and 800 mM urea. Protein was then concentrated using a 30,000 MWCO Amicon filter and subjected to size-exclusion chromatography using Äkta Pure system and a

Superose 6 column pre-equilibrated in filtration buffer (20 mM HEPES, 100 mM KCl, 800 mM urea, 1 mM DTT, pH 8). Peak fractions were flash-frozen in liquid nitrogen and stored at −80°C.

## IMPDH assembly

Filaments (or, depending on ligand state, free octamers) were prepared by diluting aliquots of purified hIMPDH2 in activity buffer (20 mM HEPES, 100 mM KCl, 1 mM DTT, pH 7.5) to 2 µM in the presence of varying concentrations of ATP, GTP, IMP, and/or $NAD^+$ and incubating for 30 min at 20°C. Nucleotide stocks were prepared using ATP disodium salt hydrate (Sigma A2383), GTP sodium salt hydrate (Sigma G8877), IMP disodium salt hydrate (Acros AC226260050), and β-Nicotinamide adenine dinucleotide hydrate (Sigma N6522), For all buffers, protein preparations, and nucleotide stocks, final pH was checked and titrated if necessary to 7.5 with KOH or HCl.

## IMPDH activity assays

Protein aliquots were diluted in activity buffer and pre-treated with varying concentrations of ATP, GTP, and IMP for 30 min at 20°C, in 96 well UV transparent plates (Corning model 3635). Reactions (100 uL total) were initiated by addition of varying concentrations of $NAD^+$. NADH production was measured by optical absorbance (340 nm) in real-time using a Varioskan Lux microplate reader (Thermo Scientific) at 20°C, 1 measurement/min, for 20 min; absorbance was correlated with NADH concentration using a standard curve. Specific activity was calculated by linear interpretation of the reaction slope for a 5 min window beginning 3 min after reaction initiation.

## Negatively stained electron microscopy

Protein preparations were applied to glow-discharged continuous carbon EM grids and negatively stained with 2% uranyl formate. Grids were imaged by transmission electron microscopy using an FEI Tecnai G2 Spirit at 120kV acceleration voltage and a Gatan Ultrascan 4000 CCD using the Leginon software package (*Suloway et al., 2009*). Micrographs were collected at a nominal 67,000x magnification (pixel size 1.6 Å). GCTF was used for contrast transfer function (CTF) estimation, and Relion for particle picking and 2D classification (*Scheres, 2012*; *Zhang, 2016*; *Zivanov et al., 2018*).

## Electron cryo-microscopy sample preparation and data collection

Protein preparations were applied to glow-discharged C-flat holey carbon EM grids (Protochips), blotted, and plunge-frozen in liquid ethane using an Vitrobot plunging apparatus (FEI) at 20°C, 100% relative humidity. High-throughput data collection was performed using an FEI Titan Krios transmission electron microscope operating at 300 kV and equipped with a Gatan image filter (GIF) and post-GIF Gatan K2 Summit direct electron detector using the Leginon software package (*Suloway et al., 2009*). For the two datasets with non-filament octamers of IMPDH, which exhibit a preferred orientation, it was necessary to collect images with the stage tilted in order to capture a sufficient range of views for 3D reconstruction (*Figure 4—figure supplement 3A*, *Figure 5—figure supplement 2A*).

## Electron cryo-microscopy image processing

Movies were collected in super-resolution mode, then aligned and corrected for beam-induced motion using Motioncor2, with 2X Fourier binning and dose compensation applied during motion correction (*Suloway et al., 2009*; *Zheng et al., 2017*). CTF was estimated using GCTF (*Zhang, 2016*). Relion 3.0 was used for all subsequent image processing (*Zhang, 2016*; *Zivanov et al., 2018*). Although multiple datasets of hIMPDH2 under the different ligand states were collected, each dataset was individually processed using approximately the same overall pipeline (*Figure 2—figure supplement 1E–G*), with some variations from dataset to dataset (*Figure 3—figure supplement 1*, *Figure 4—figure supplement 2*, *Figure 4—figure supplement 3*, *Figure 5—figure supplement 2*, *Figure 5—figure supplement 3*, *Figure 5—figure supplement 4*). First, for each dataset, autopicking templates and initial 3D references maps were prepared by manually picking and extracting boxed particles from a small subset of micrographs, and classifying/refining in 2D and 3D. For these initial 3D refinements, a featureless, soft-edged cylinder was used as a refinement template of filaments, and a previously published cryo-em map (EMDB-8692) was used as template for non-filament octamers (*Anthony et al., 2017*). Because IMPDH filament segments possess D4

point-group symmetry, two different locations along filaments may be used as symmetry origins: the centers of canonical octamer segments, or the centers of the assembly interface between segments. For the filament datasets, we prepared and used auto-picking templates centered on the filament assembly interface. For the datasets containing non-filament octamers of hIMPDH2, auto-picking templates centered on these non-filament particles were also included. Due to the flexibility of hIMPDH2 filaments, helical segments were processed as single particles, and at no point was helical symmetry applied during image processing. After template-based autopicking of each complete dataset, picked particles were boxed and extracted from micrographs, and subjected to hierarchical 2D classification to select the best-resolved classes. These selected particles were then auto-refined in 3D as a single class with symmetry applied (D4 for filament segments and free octamers, C4 for filament ends). Exploratory image processing of the assembly interface-centered filament reconstructions made it apparent that the eight catalytic domains surrounding this interface appeared conformationally homogenous, while the Bateman domains and neighboring octamers appeared conformationally varied. Additionally, due to the flexibility of the filaments, and the tendency of filament ends to adhere to the air-water interface, many filaments were tilted out of plane, with neighboring segments overlapping in projection.

To improve resolution, partial signal subtraction was performed at this stage, using a mask that left only the central eight catalytic domains of the filament assembly interface, subtracting the poorly resolved Bateman domains and neighboring segments, which served to improve resolution after subsequent auto-refinement. Per-particle defocus and per-micrograph astigmatism were then optimized using CTF refinement, which improved resolution further. The resulting consensus refinements of the filament assembly interfaces were well-resolved, however data on Bateman domain conformation was missing, with these regions very poorly resolved when subtracted regions were restored to the reconstructions by reversion to original non-subtracted particles (data not shown). To resolve the different Bateman domain conformations, we applied particle symmetry expansion (D4 to C1) and classified particles without additional alignment. Because at this stage the reconstructions were centered on the filament assembly interface, each boxed 'particle' contained elements of two different neighboring octamers. The potential conformational space was reduced by applying a mask enclosing only one of these two octamers. By hierarchical focused classification of the off-origin octamers we were able to classify multiple conformations of the octameric filament segments, as well as incomplete segments and filament ends. Symmetry expansion was also applied to the non-filament octamer datasets, with a mask including the entire particle, which allowed classification of the most symmetric and well-resolved classes. To further improve resolution of the varying symmetry-expanded segment classes, the reconstruction symmetry origins were moved from the filament assembly interface to the canonical octamers by re-extraction with re-centering. For each class, symmetry was then collapsed by removing redundant overlapping particles, Euler angles reset to zero. After auto-refining once again, we observed that the most well-resolved octameric segments from the asymmetric symmetry-expanded classifications exhibited some apparent symmetry, with fully extended or fully compressed octamers appearing D4 symmetric, and some bent classes apparently D1 symmetric. We therefore applied these symmetries during subsequent refinement and classification of these new octamer-centered classes. As before, signal subtraction of neighboring filament segments improved resolution considerably. Additional rounds of CTF refinement and 3D classification identified the best-resolved particles from each of these conformational classes. Final overall resolution (according to the FSC = 0.143 criterion), as well as local resolution, was assessed using Relion postprocessing.

## Model building and refinement

As initial templates for model building, two hybrid models (representing hIMPDH2 in either an extended or compressed state) were prepared by combining elements from existing crystal structures. For both templates, the catalytic domain and substrate poses (residues 18–107, and 245–514) were taken from a crystal structure of an inhibitor-bound hIMPDH2 (PDB 1nf7), and the Bateman domains and ligand poses (residues 108–244) were based on fungal (*A. gossypii)* IMPDH crystallized in either the extended or compressed states (PDB 5mcp and 5tc3, respectively), and SWISS-MODEL homology modeling (*Buey et al., 2017*; *Sintchak et al., 1996*; *Waterhouse et al., 2018*). The N-terminus (residues 1–17) were modeled by hand. In all maps, a single active site loop (residues 421 to 436) was unresolved, and these residues were not modeled. After rigid-body fitting of templates

into the cryo-EM densities using UCSF Chimera, repeated cycles of manual fitting with Coot, automated fitting with phenix.real_space_refine (employing rigid-body refinement, NCS constraints, gradient-driven minimization and simulated annealing) and local B-factor sharpening of cryo-EM data via LocScale were used for final atomic model refinement and local sharpening of cryo-EM maps (*Adams et al., 2012*; *Emsley et al., 2010*; *Jakobi et al., 2017*; *Pettersen et al., 2004*). Final models were evaluated with MOLPROBITY and EMRinger (*Barad et al., 2015*; *Chen et al., 2010*). Data collection parameters and refinement statistics are summarized in *Figure 3—source data 1*, *Figure 3—source data 2*, *Figure 3—source data 3*, *Figure 4—source data 1*, *Figure 4—source data 2*, *Figure 5—source data 1*. Figures were prepared with UCSF Chimera (*Pettersen et al., 2004*). Fourier shell correlations between the final models and maps were calculated with phenix.mtriage, and to probe for overfitting, a model refined against one half-map was likewise used to calculate FSCs between both that half-map (FSC-work) and the other half-map (FSC-test) (*Afonine et al., 2018*) (*Figure 3—figure supplement 2*, *Figure 4—figure supplement 3*, *Figure 4—figure supplement 5*, *Figure 5—figure supplement 5*, *Figure 5—figure supplement 6*).

## Classification of cryo-EM data against synthetic templates and calculation of interfaced octamer odds ratios

The five interface-centered filament particle stacks were re-extracted to a larger box size and a common pixel size (2 Å/px) and a larger box size (576 Å) that completely included both interfaced octamers (*Figure 5—figure supplement 8A*). Synthetic templates representing different octamer conformations and orientations for either of these interfaced octamers were prepared using a subset of the final experimental models, and simulating EM density using the Eman2 tool pdb2mrc (*Tang et al., 2007*) (*Figure 5—figure supplement 8B*). The paired octamers from each interface were then classified in Relion for a single iteration, without additional alignment, using masks to focus on one octamer or the other. Odds ratios (and associated 95% confidence intervals) were then calculated for all possible pairs of classified conformations between the sets of interfaced octamers ((*Figure 5—figure supplement 8C*, (*Figure 5—figure supplements 9–13*).

## Acknowledgements

The authors thank the Arnold and Mabel Beckman Cryo-EM Center at the University of Washington for electron microscope use. We are grateful to J Quispe for technical support and advice. We thank A Burrell, G Cai, A Horowitz, K Hvorecny, and E Lynch for valuable feedback. This work was supported by the US National Institutes of Health (R01 GM118396 to JMK).

## Additional information

### Funding

| Funder | Grant reference number | Author |
|---|---|---|
| National Institutes of Health | 5R01GM118396-04 | Justin M Kollman |

The funders had no role in study design, data collection and interpretation, or the decision to submit the work for publication.

### Author contributions

Matthew C Johnson, Conceptualization, Data curation, Formal analysis, Validation, Investigation, Visualization, Methodology; Justin M Kollman, Conceptualization, Resources, Supervision, Funding acquisition, Project administration

### Author ORCIDs

Matthew C Johnson http://orcid.org/0000-0002-1477-7801
Justin M Kollman https://orcid.org/0000-0002-0350-5827

### Decision letter and Author response

Decision letter https://doi.org/10.7554/eLife.53243.sa1

Author response https://doi.org/10.7554/eLife.53243.sa2

## Additional files

### Supplementary files

• Transparent reporting form

### Data availability

The cryo-EM maps described here have been deposited in the Electron Microscopy Data Bank with accession numbers 20687, 20688, 20690, 20691, 20701, 20704, 20705, 20706, 20707, 20709, 20716, 20718, 20720, 20722, 20723, 20725, 20742, 20741, and 20743. The refined atomic coordinates have been deposited in the Protein Data Bank with accession numbers 6U8E, 6U8N, 6U8R, 6U8S, 6U9O, 6UA2, 6UA4, 6UA5, 6UAJ, 6UC2, 6UDP, 6UDO, and 6UDQ.

The following datasets were generated:

| Author(s) | Year | Dataset title | Dataset URL | Database and Identifier |
|---|---|---|---|---|
| Johnson MC, Kollman JM | 2019 | IMPDH2 ATP, IMP, NAD+ assembly interface cryo-EM map | https://www.ebi.ac.uk/pdbe/entry/emdb/EMD-20687 | Electron Microscopy Data Bank, 20687 |
| Johnson MC, Kollman JM | 2019 | IMPDH2 ATP, IMP, NAD+ assembly interface atomic model | https://www.ebi.ac.uk/pdbe/entry/pdb/6u8e | Protein Data Bank, 6U8E |
| Johnson MC, Kollman JM | 2019 | IMPDH2 ATP, IMP, NAD+ fully extended cryo-EM map | https://www.ebi.ac.uk/pdbe/entry/emdb/EMD-20688 | Electron Microscopy Data Bank, 20688 |
| Johnson MC, Kollman JM | 2019 | IMPDH2 ATP, IMP, NAD+ fully extended atomic model | https://www.ebi.ac.uk/pdbe/entry/pdb/6u8n | Protein Data Bank, 6U8N |
| Johnson MC, Kollman JM | 2019 | IMPDH2 ATP, IMP, NAD+ Bent (1:3) cryo-EM map | https://www.ebi.ac.uk/pdbe/entry/emdb/EMD-20690 | Electron Microscopy Data Bank, 20690 |
| Johnson MC, Kollman JM | 2019 | IMPDH2 ATP, IMP, NAD+ Bent (1:3) atomic model | https://www.ebi.ac.uk/pdbe/entry/pdb/6u8r | Protein Data Bank, 6U8R |
| Johnson MC, Kollman JM | 2019 | IMPDH2 ATP, NAD+ assembly interface cryo-EM map | https://www.ebi.ac.uk/pdbe/entry/emdb/EMD-20718 | Electron Microscopy Data Bank, 20718 |
| Johnson MC, Kollman JM | 2019 | IMPDH2 ATP, NAD+ fully extended cryo-EM map | https://www.ebi.ac.uk/pdbe/entry/emdb/EMD-20716 | Electron Microscopy Data Bank, 20716 |
| Johnson MC, Kollman JM | 2019 | IMPDH2 ATP, NAD+ fully compressed cryo-EM map | https://www.ebi.ac.uk/pdbe/entry/emdb/EMD-20709 | Electron Microscopy Data Bank, 20709 |
| Johnson MC, Kollman JM | 2019 | IMPDH2 ATP, IMP assembly interface cryo-EM map | https://www.ebi.ac.uk/pdbe/entry/emdb/EMD-20723 | Electron Microscopy Data Bank, 20723 |
| Johnson MC, Kollman JM | 2019 | IMPDH2 ATP, IMP fully extended cryo-EM map | https://www.ebi.ac.uk/pdbe/entry/emdb/EMD-20722 | Electron Microscopy Data Bank, 20722 |
| Johnson MC, Kollman JM | 2019 | IMPDH2 ATP, IMP fully compressed cryo-EM map | https://www.ebi.ac.uk/pdbe/entry/emdb/EMD-20720 | Electron Microscopy Data Bank, 20720 |
| Johnson MC, Kollman JM | 2019 | IMPDH2 ATP, GTP free octamer | https://www.ebi.ac.uk/pdbe/entry/emdb/EMD-20725 | Electron Microscopy Data Bank, 20725 |
| Johnson MC, Kollman JM | 2019 | IMPDH2 ATP, GTP free octamer | https://www.ebi.ac.uk/pdbe/entry/pdb/6uc2 | Protein Data Bank, 6UC2 |
| Johnson MC, Kollman JM | 2019 | IMPDH2 ATP, GTP, IMP assembly interface | https://www.ebi.ac.uk/pdbe/entry/emdb/EMD-20742 | Electron Microscopy Data Bank, 20742 |
| Johnson MC, Koll- | 2019 | IMPDH2 ATP, GTP, IMP assembly | https://www.ebi.ac.uk/ | Protein Data Bank, |

| | | | | | |
|---|---|---|---|---|---|
| man JM | | | interface | pdbe/entry/pdb/6udp | 6UDP |
| Johnson MC, Koll-man JM | | 2019 | IMPDH2 ATP, GTP, IMP fully compressed | https://www.ebi.ac.uk/pdbe/entry/emdb/EMD-20741 | Electron Microscopy Data Bank, 20741 |
| Johnson MC, Koll-man JM | | 2019 | IMPDH2 ATP, GTP, IMP fully compressed | https://www.ebi.ac.uk/pdbe/entry/pdb/6udo | Protein Data Bank, 6UDO |
| Johnson MC, Koll-man JM | | 2019 | IMPDH2 ATP, GTP, IMP fully compressed end | https://www.ebi.ac.uk/pdbe/entry/emdb/EMD-20743 | Electron Microscopy Data Bank, 20743 |
| Johnson MC, Koll-man JM | | 2019 | IMPDH2 ATP, GTP, IMP fully compressed end | https://www.ebi.ac.uk/pdbe/entry/pdb/6udq | Protein Data Bank, 6UDQ |
| Johnson MC, Koll-man JM | | 2019 | IMPDH2 ATP, GTP, IMP, NAD+ assembly interface cryo-EM map | https://www.ebi.ac.uk/pdbe/entry/emdb/EMD-20691 | Electron Microscopy Data Bank, 20691 |
| Johnson MC, Koll-man JM | | 2019 | IMPDH2 ATP, GTP, IMP, NAD+ assembly interface atomic model | https://www.ebi.ac.uk/pdbe/entry/pdb/6u8s | Protein Data Bank, 6U8S |
| Johnson MC, Koll-man JM | | 2019 | IMPDH2 ATP, GTP, IMP, NAD+ bent (2:2) cryo-EM map | https://www.ebi.ac.uk/pdbe/entry/emdb/EMD-20704 | Electron Microscopy Data Bank, 20704 |
| Johnson MC, Koll-man JM | | 2019 | IMPDH2 ATP, GTP, IMP, NAD+ bent (2:2) atomic model | https://www.ebi.ac.uk/pdbe/entry/pdb/6ua2 | Protein Data Bank, 6UA2 |
| Johnson MC, Koll-man JM | | 2019 | IMPDH2 ATP, GTP, IMP, NAD+ bent (3:1) cryo-EM map | https://www.ebi.ac.uk/pdbe/entry/emdb/EMD-20705 | Electron Microscopy Data Bank, 20705 |
| Johnson MC, Koll-man JM | | 2019 | IMPDH2 ATP, GTP, IMP, NAD+ bent (3:1) atomic model | https://www.ebi.ac.uk/pdbe/entry/pdb/6ua4 | Protein Data Bank, 6UA4 |
| Johnson MC, Koll-man JM | | 2019 | IMPDH2 ATP, GTP, IMP, NAD+ fully compressed cryo-EM map | https://www.ebi.ac.uk/pdbe/entry/emdb/EMD-20701 | Electron Microscopy Data Bank, 20701 |
| Johnson MC, Koll-man JM | | 2019 | IMPDH2 ATP, GTP, IMP, NAD+ fully compressed atomic model | https://www.ebi.ac.uk/pdbe/entry/pdb/6u9O | Protein Data Bank, 6U9O |
| Johnson MC, Koll-man JM | | 2019 | IMPDH2 ATP, GTP, IMP, NAD+ free octamer cryo-EM map | https://www.ebi.ac.uk/pdbe/entry/emdb/EMD-20707 | Electron Microscopy Data Bank, 20707 |
| Johnson MC, Koll-man JM | | 2019 | IMPDH2 ATP, GTP, IMP, NAD+ free octamer atomic model | https://www.ebi.ac.uk/pdbe/entry/pdb/6uaj | Protein Data Bank, 6UAJ |
| Johnson MC, Koll-man JM | | 2019 | IMPDH2 ATP, GTP, IMP, NAD+ free interfacial octamer cryo-EM map | https://www.ebi.ac.uk/pdbe/entry/emdb/EMD-20706 | Electron Microscopy Data Bank, 20706 |
| Johnson MC, Koll-man JM | | 2019 | IMPDH2 ATP, GTP, IMP, NAD+ free interfacial octamer atomic model | https://www.ebi.ac.uk/pdbe/entry/pdb/6ua5 | Protein Data Bank, 6UA5 |

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
