## [Decision Letter]

**Acceptance summary:**

Human IMPDH2 forms filaments and other higher order structures when the enzyme inhibited and guanine nucleotide pools are depressed. Curiously, this structural transformation appears to have very little effect on enzymatic activity. This paper reports a series of EM structures of filaments formed under different ligand conditions, revealing several different conformational states. Based on these structures, the authors propose an elegant mechanism of allosteric regulation where filament formation regulates GTP feedback inhibition that may at last explain the mysterious physiological role of filament formation. This is an intriguing and potentially very important contribution. It was noted that IMPDH is a major target of immunosuppressive drugs such as mycophenolic acid. As such, a deeper structural understanding of its regulation has implications for further refinements to this class of drugs.

**Decision letter after peer review:**

Thank you for submitting your article "Ensemble cryo-EM structures demonstrate human IMPDH2 filament assembly tunes allosteric regulation" for consideration by *eLife*. Your article has been reviewed by three peer reviewers, including Sjors HW Scheres as the Reviewing Editor and Reviewer #1, and the evaluation has been overseen by Cynthia Wolberger as the Senior Editor.

The reviewers have discussed the reviews with one another and the Reviewing Editor has drafted this decision to help you prepare a revised submission.

Essential revisions:

1) The enzyme preparation appears to be much less active (by a factor of 20) than those reported from other laboratories. The value of kcat is typically in approx. 1 s^-1^ (see BRENDA website). Figure 2D reports a specific activity of approx. 50 nmoles NADH produced per min per mg of enzyme. Converting (please check the math):

(50 nmoles min-1mg-1) divided by 60 s min-1 = ~0.9 nmoles s^-1^ mg-1

1 x 10-3 g IMPDH divided 55,000 g/mol = 18 nmoles

0.9 nmole s^-1^ mg -1 divided 18 nmoles mg-1 = ~ 0.05 s^-1^

If only 5% of the enzyme is active, then the relevance of these structures is in doubt. It is possible that this discrepancy arises from issues with the assay itself rather than the integrity of the enzyme preparation. The description of the assay does not include the enzyme concentration. However, samples were taken directly from the assay for structural work, which suggests that the concentration in the assay may have been 2 uM. This is a very high concentration. If the enzyme were as active as observed in other laboratories, the rate of NADH production would be:

2 μm x 1 s^-1^ x 60 s/min = 120 μm min-1

The starting substrate concentrations were 2 mM- all the substrate would be consumed during the 20 min assay. The rates were measured starting 3 min after initiation (unclear why), at a point where approximately 20% of the substrates would have been consumed, so the reaction is no longer under initial rate conditions. The resulting products would be expected to inhibit the reaction. Note that NAD inhibits at high concentrations, adding another complication. So it is quite possible that the assays did not accurately measure enzyme activity.

2) Another concern is the potential for inadvertent changes in reaction conditions at high nucleotide concentrations. The buffer contains 100 mM KCl and only 20 mM Hepes, and pH 7 is at the lower end of its useful range. Concentrations of nucleotides go as high as 20 mM in the experiments, which could change pH and also affect ionic strength. Was the pH checked?

3) The authors do not describe the reagents- it is important to include which salts of IMP, ATP, etc. are used. In particular, the behavior of ATP can change markedly depending on whether Na or Mg salts were used. The enzyme is activated by monovalent cations, and the monovalent cation binding site lies at or near the octamer interface. Changes in the nature of the monovalent cation could also change the octamer interface. This is a question for another day, but it would be good to know what monovalent cations are present in the reactions. For example, the disodium ATP salt is one of the most commonly used ATP reagents- at 20 mM ATP, the sodium concentration would be 40 mM. The Km for Na is similar to K, so approx. 30% monomers would contain Na instead of K.

4) The regulation of metabolic enzyme polymerization appears to act, primarily, by altering the affinity of the "monomer" for the end of the polymer. Basically, small molecule binding to the monomer shifts the equilibrium distribution of the protein between the monomer/filament states. The effect of GTP on the filament curvature suggests that it is incorporating into the filament, altering the structure and destabilizing it. This could have two roles: 1) if the filament strain is sufficient, the filament could break. This would accelerate disassembly by increasing the number of ends available for depolymerization since this is an end limited process or 2) the only relevant strain is at the ends, where is accelerates subunit loss from the filament. While these are not mutually exclusive, it should be possible to tease these possibilities apart by looking at the length distribution of filaments at very low GTP concentration and/or short time points. Given the ensemble nature of IMPDH filaments and the fact that the authors have some of the titration data already, it shouldn't be too burdensome to take a look at the role of the strained conformation in promoting disassembly. It could be that the authors already have sufficient data at hand to address this based on the fact that the filaments are more resistant to the effects of GTP than the monomer arguing that the effects are at the level of the filament. Further clarification of this would be helpful.

5) One could analyse the orientations and classes of individual particle images along each filament to get an idea about the extended/compressed conformations of the Bateman domains in subsequent octamers inside each filaments. This could lead to insights whether these conformations are directly related to the filaments bending in a specific direction, e.g. are compressed conformations on the inside of bends? Was such an analysis attempted?

6) Despite atomic-model refinements, no FSCs between models and maps are presented. This should be remediated in a revised version. Three curves should be shown for each map in which an atomic model was refined: a) between the final map and the final model; b) between a model refined in one of the half-maps against that same half-map (FSC-work); and c) of the same model from b) against the other half-map (FSC-test). A large difference between FSC-work and FSC-test should be avoided (by adjusting weights in refinement), as they would be an indication of overfitting.

---

## [Author Response]

Essential revisions:1) The enzyme preparation appears to be much less active (by a factor of 20) than those reported from other laboratories. The value of kcat is typically in approx. 1 s^-1^ (see BRENDA website). Figure 2D reports a specific activity of approx. 50 nmoles NADH produced per min per mg of enzyme. Converting (please check the math):(50 nmoles min-1mg-1) divided by 60 s min-1 = ~0.9 nmoles s^-1^ mg-11 x 10-3 g IMPDH divided 55,000 g/mol = 18 nmoles0.9 nmole s^-1^ mg -1 divided 18 nmoles mg-1 = ~ 0.05 s^-1^If only 5% of the enzyme is active, then the relevance of these structures is in doubt. It is possible that this discrepancy arises from issues with the assay itself rather than the integrity of the enzyme preparation. The description of the assay does not include the enzyme concentration. However, samples were taken directly from the assay for structural work, which suggests that the concentration in the assay may have been 2 uM. This is a very high concentration. If the enzyme were as active as observed in other laboratories, the rate of NADH production would be:2 μm x 1 s^-1^ x 60 s/min = 120 μm min-1The starting substrate concentrations were 2 mM- all the substrate would be consumed during the 20 min assay. The rates were measured starting 3 min after initiation (unclear why), at a point where approximately 20% of the substrates would have been consumed, so the reaction is no longer under initial rate conditions. The resulting products would be expected to inhibit the reaction. Note that NAD inhibits at high concentrations, adding another complication. So it is quite possible that the assays did not accurately measure enzyme activity.

For our kinetic assays, the protein concentration was, in fact, 2 uM, and we now note this in the Materials and methods section. This protein concentration was selected because it was the optimum for cryo-EM particle dispersion, and we felt it necessary that our kinetic experiments be performed under identical conditions as our EM experiments. The timeframe in which we measure reaction velocities is in the linear range of the activity assays, and we now show representative traces of NADH generated vs. time in Figure 2—figure supplement 2A,B for a range of protein concentrations.

Indeed, the activity levels we report are lower than some previous studies have found for human IMPDH2. One major consideration is temperature – we perform our experiments at 24**°**C, while most others have reported activities at 37**°**C. We see an ~3-fold increase in activity at 37**°**C. We also see a minor effect of enzyme concentration, with higher concentrations having a slightly lower activity (data now included as Figure 2 —figure supplement 2C). At the higher temperature, our kcat is similar to the most recently reported literature value of 0.18 s^-1^ (Fernández-Justel et al., 2019). Importantly, that study and ours were similar in working at high enzyme concentrations (1.8 and 2 uM). Moreover, these are the only two studies we are aware of that have included careful biophysical characterization of the enzyme used in biochemical assays (SAXS and negative stain EM in Fernández-Justel, et al. and cryo-EM in our paper); this is likely important as, in our hands at least, non-specific aggregation of IMPDH2 is a common problem that might have influenced activity in other studies.

2) Another concern is the potential for inadvertent changes in reaction conditions at high nucleotide concentrations. The buffer contains 100 mM KCl and only 20 mM Hepes, and pH 7 is at the lower end of its useful range. Concentrations of nucleotides go as high as 20 mM in the experiments, which could change pH and also affect ionic strength. Was the pH checked?

In responding to this comment we realized an error in the manuscript: the final pH for all kinetic assays and EM preparations was 7.5, not 7; the revised manuscript contains this correction. The final pH for all reagents, including buffers, protein preparations, and nucleotide stocks, was adjusted if necessary to pH 7.5 using KOH or HCl.

3) The authors do not describe the reagents- it is important to include which salts of IMP, ATP, etc. are used. In particular, the behavior of ATP can change markedly depending on whether Na or Mg salts were used. The enzyme is activated by monovalent cations, and the monovalent cation binding site lies at or near the octamer interface. Changes in the nature of the monovalent cation could also change the octamer interface. This is a question for another day, but it would be good to know what monovalent cations are present in the reactions. For example, the disodium ATP salt is one of the most commonly used ATP reagents- at 20 mM ATP, the sodium concentration would be 40 mM. The Km for Na is similar to K, so approx. 30% monomers would contain Na instead of K.

Sodium salts of ATP, GTP, and IMP were, in fact, used for these experiments. Specifically: ATP disodium salt hydrate (Sigma A2383), GTP sodium salt hydrate (Sigma G8877) and IMP disodium salt hydrate (Acros AC226260050). This information is included in the revised manuscript.

4) The regulation of metabolic enzyme polymerization appears to act, primarily, by altering the affinity of the "monomer" for the end of the polymer. Basically, small molecule binding to the monomer shifts the equilibrium distribution of the protein between the monomer/filament states. The effect of GTP on the filament curvature suggests that it is incorporating into the filament, altering the structure and destabilizing it. This could have two roles: 1) if the filament strain is sufficient, the filament could break. This would accelerate disassembly by increasing the number of ends available for depolymerization since this is an end limited process or 2) the only relevant strain is at the ends, where is accelerates subunit loss from the filament. While these are not mutually exclusive, it should be possible to tease these possibilities apart by looking at the length distribution of filaments at very low GTP concentration and/or short time points. Given the ensemble nature of IMPDH filaments and the fact that the authors have some of the titration data already, it shouldn't be too burdensome to take a look at the role of the strained conformation in promoting disassembly. It could be that the authors already have sufficient data at hand to address this based on the fact that the filaments are more resistant to the effects of GTP than the monomer arguing that the effects are at the level of the filament. Further clarification of this would be helpful.

We also were curious whether the strain that induces disassembly occurs only at the filament ends, or if depolymerization can occur at any filament interface. However, we were unsuccessful in designing an experiment to address this. As explained below in more detail in our response to comment #5, we performed calculations based on our cryo-EM data to probe whether the compressed filament conformations were more, or less, likely to be near filament ends, but our results were inconclusive. Although the GTP titration data we present in our manuscript does include low concentrations of GTP, because these samples were prepared ~30 minutes after GTP treatment any assembly/disassembly of filaments is presumed to be at equilibrium. In such a case, we would not expect the distributions of filament lengths between our two possible scenarios to be different. In theory, it should be possible to obtain the necessary data from non-equilibrium states, by preparing EM grids at different time points after GTP treatment. We have previously published results of such an experiment, in which we found that the time-scale of GTP-induced IMPDH2 filament disassembly was on the order of seconds (Duong-Ly et al., 2018). Unfortunately, given this rate of disassembly, and the technical limitations of EM grid preparation, we were not able to sample sufficient time points to obtain the data needed. We are thus forced to leave this question for future experiments.

5) One could analyse the orientations and classes of individual particle images along each filament to get an idea about the extended/compressed conformations of the Bateman domains in subsequent octamers inside each filaments. This could lead to insights whether these conformations are directly related to the filaments bending in a specific direction, e.g. are compressed conformations on the inside of bends? Was such an analysis attempted?

In response to this comment, we have performed additional analysis, classifying the filament interface centered cryo-EM data against a series of synthetic templates. These interface centered particle stacks are each the consensus of all filament segments from each dataset, and each interface-centered “particle” contains two interfaced octamers. We independently classified each of the two octamers against simulated EM density templates prepared from the final atomic models. To check for correlation of conformations across the interface, we compared the classified particles using pairwise odds ratios. We have included this new analysis in the revised manuscript as Figure 5—figure supplement 8-13, and have added new passages to the Results section describing it. This analysis provides a large number of pairwise comparisons across all of our datasets, but the overall picture is one of minimal to no correlation between the conformational states of adjacent octamers in a filament.

6) Despite atomic-model refinements, no FSCs between models and maps are presented. This should be remediated in a revised version. Three curves should be shown for each map in which an atomic model was refined: a) between the final map and the final model; b) between a model refined in one of the half-maps against that same half-map (FSC-work); and c) of the same model from b) against the other half-map (FSC-test). A large difference between FSC-work and FSC-test should be avoided (by adjusting weights in refinement) as they would be an indication of overfitting.

The three types of model-map FSCs requested by the reviewer (final model-map FSC, FSC-work, and FSC-test) have been prepared for all final models and are included in the revised manuscript as Figure 3—figure supplement 2, Figure 4—figure supplement 3, Figure 4—figure supplement 5, Figure 5—figure supplement 5, and Figure 5—figure supplement 6.